

# Quantification of runoff generation from a combined glacier and *páramo* catchment within an Ecological Reserve in the Ecuadorian highlands

Verónica Minaya[a,b], Vivian Camacho Suarez[c], Jochen Wenninger[a,b], Arthur Mynett[a,b]

    a.    Department of Water Science and Engineering, UNESCO-IHE Institute for Water Education, P.O. Box 3015, 2601 DA Delft, the Netherlands

    b.    Section of Water Resources, Delft University of Technology, P.O. Box 5048, 2600 GA Delft, the Netherlands

    c.    University of Sheffield, Civil Engineering Department, Sheffield, UK

*Correspondence to*: Veronica Minaya (verominaya81@hotmail.com)

**Abstract.**

Hydrological processes in combined glacier and *páramo* catchments are vitally important to serve the water needs of communities in the surrounding areas. Previous studies have shown that the melting of glaciers contributes to runoff generation and that the *páramo* ecosystem acts as a natural sponge, which plays an important role in regulating the runoff during the dry-season. However, not all runoff processes are well-understood in the Andean region due to the high spatial variability of precipitation, young volcanic ash soil properties, soil moisture dynamics and other local factors such as vegetation interception and high radiation that might influence the hydrological behaviour. In addition, there is a lack of evidence of the origin and quantification of the contribution of runoff components in the *páramo* ecosystem. This study focuses on data collection and experimental investigations in a small catchment (15.2 km$^2$) within an ecological reserve in the Ecuadorian Andean region. The approach consists of the identification of suitable environmental tracers and hydrochemical features to identify the various runoff sources in order to determine their respective contribution during dry and wet conditions. The results show the great importance of the *páramo* on the contribution to total runoff during baseflow and rainfall conditions.

**Key words:** environmental tracers, runoff generation, water chemistry, *páramos,* tropical grasslands

## 1.      Introduction

Tropical grasslands are one of the most abundant but probably least-understood ecosystems in terms of their biological and physical processes. In the Andean region such grasslands are known as *páramos* and they have been recognized for their importance in providing water for agriculture and urban use (Buytaert and Beven, 2011), and sustaining biodiversity and unique ecological processes (Hofstede et al., 2002; Madriñán et al., 2013). The importance of the *páramos* is associated with the tremendous capacity of water retention in its volcanic ash soil covered by vegetation (Poulenard et al., 2002; Roa-García et al., 2011). Ecuador has nearly 12,500 km$^2$ of *páramo* of which 64% in areas above 3000 m a.s.l. has been transformed or degraded (Hofstede et al., 2002) and the remaining areas are currently under constant pressure. At a higher altitude (>4000 m a.s.l.), these ecosystems are influenced by permanent snow and glaciers that feed directly the river's drainage system or may contribute further downstream by resurgence (Cauvy-Frauník et al., 2013; Favier et al., 2008; Villacis, 2008).

A lot of attention has been put on the relationship between climate change and retreating glaciers globally (Beniston, 2003), particularly because climate change in conjunction with the rapid change in land use can jeopardize the water quantity and quality of the *páramos* (Buytaert and Beven, 2009; Buytaert et al., 2006b; Jansky et al., 2002). Bradley *et al.* (2006) showed clear evidence of faster surface temperature changes in higher elevations compared to lower elevations in the Tropical Andes with a rate of 0.11°C per decade in the period 1939 to 1998. The concern of the scientific community lies in the implications of



the melting water and its impact on hydrological systems alongside the response of the terrestrial, aquatic biota (Cauvy-Frauníe et al., 2013) and water security for communities that rely on these catchments in the tropical regions (Brown et al., 2010; Kaser et al., 2010). The complexity of these glacierized-*páramo* catchments is higher than those in temperate regions since *páramo* catchments are more affected by a continuous ablation at all altitudes (Kaser and Osmaston, 2002) leading to changes in the

hydrological, geomorphic and ecological processes. Due to the orographic properties of these high mountainous regions in the *páramos*, the precipitation regime has a remarkably large spatial variability (Buytaert et al., 2006b; Celleri and Feyen, 2009). Several studies enhance the importance of a fair understanding of the hydrological complexity of these interconnected systems and the implications for water resources management in the region (Buytaert and Beven, 2011; Buytaert et al., 2010; Carrillo-Rojas et al., 2016; Cuesta et al., 2013; Viviroli et al., 2011). Tracer experiments have been widely-used to provide more

information about the connectivity and time scales of the contribution of the main runoff sources and flow pathways to total runoff (Condom et al., 2012; Dahlke et al., 2012; Huss et al., 2008; Munyaneza et al., 2012; Villacis et al., 2008; Wenninger et al., 2008; Windhorst et al., 2013). However, a suitable spatial hydrochemical characterization and quantification of the different contributions from groundwater reservoirs and meltwater infiltrations in these catchments of complex geology and topography remains a challenge.

In this regard, this study aims to provide effective tools to investigate the origin of the main runoff components and to quantify their contributions using environmental tracers (isotopes and major ions). In addition, this study comprises a complete hydrochemical analysis of the runoff components separated by source and location. This will provide a fair understanding of the hydrological interactions of a glacierized-*páramo* system.

## 2.   Study Area

Location

The catchment Los Crespos-Humboldt (15.2 km$^2$) is located at the south-western slope of the Antisana volcano (0°30'S,

78°11'W) and it lies within the Antisana Ecological Reserve (628.1 km$^2$) in the Andean region of Ecuador (Figure 1a). This catchment is one of several water sources for La Mica reservoir that supplies water for the southern part of Quito, the capital city of Ecuador, located 50 km north of this catchment.

The catchment elevation ranges from 4000 to 5700 m a.s.l. and it consists of 15% glaciers, 68% *páramo* grasslands and 17% moraines (Figure 1b). The latter one is an ecosystem in transition between the glacier and the *páramo*. The *páramo* vegetation

is dominated by tussock grasses (TU) (*Calamagrostis intermedia*), acaulescent rosettes (AR) (*Werneria nubigena, Hypochaeris sessiliflora*) and cushions (CU) (*Azorella Pedunculata*) (Minaya et al., 2015) (Figure 1c). The *páramo* vegetation has adapted to specific climatic conditions of low atmospheric pressure, high radiation and wind drying effects (Luteyn, 1999). The glacier is an icecap that has retreated around 200 meters in the last 20 years (Cáceres et al., 2005; Hall et al., 2012).

Soils are mainly andosols, based on the FAO classification (Gardi et al., 2014), derived from volcanic material characterized by

their high soil moisture (Buytaert et al., 2005a) and water retention capacity (Janeau et al., 2015; Roa-García et al., 2011). In addition, studies in the area describe an elevated amount of organic carbon and mineralogical composition in the soil of the *páramos* at lower catchments, which decreases with altitudinal gradient (Minaya et al., 2015). For low and mid altitudinal ranges (4000 to 4400 m a.s.l.), the ratio of sand:silt:clay for the vegetated soils is 22:62:16, corresponding to a silty loam soil and for higher altitudes where vegetation is sparse it is 58:20:22 corresponding to a sandy clay loam soil (Minaya et al., 2015).

The soil texture influences the ecological and hydrological processes. Sandy soils drain well and reduce the capability of holding moisture; silty soils offer a high water-holding capacity. The slopes are moderate (up to 15°) in the low and mid catchment and increases up to 30° close to the moraines at higher elevations.





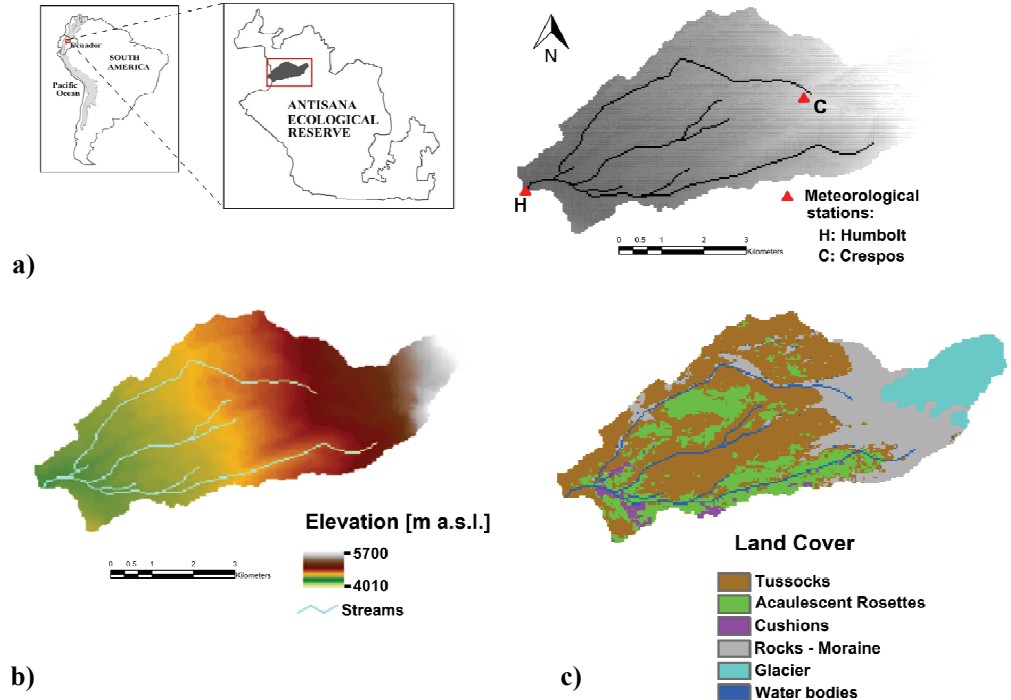

**Figure 1.** (a) Location of the Los Crespos-Humboldt basin within The Antisana Ecological Reserve, Ecuador; ( b) Elevation of the catchment with the main stream network, **(c)** Land cover map

Climate

From lower to higher altitudes the precipitation in the area increases from 900 to 1200 mm yr$^{-1}$, with an average annual precipitation of 745 mm yr$^{-1}$ to 993 mm yr$^{-1}$ for the Humboldt (4010 m a.s.l.) and the Crespos (4785 m a.s.l.) weather stations, respectively. The average temperature ranged from 7$^\circ$C to 4.8 $^\circ$C (for the period 2000 to 2011) for the same altitudes. Figure 2 shows the climate diagram of monthly average values of precipitation and maximum and minimum temperatures at the two weather stations (Humboldt and Crespos) for the period 2000 to 2011. The wet period extends typically from April to June. In

the Ecuadorian Andes, the *páramos* above 3000 m a.s.l. receive 16% more precipitation compared to other *páramos* located in the inter-Andean valleys (Buytaert and Beven, 2011). There are two main sources of precipitation: those influenced by the air masses from the Amazon region and those influenced by the inter-Andean valley regime (Vuille et al., 2000).

Precipitation has a large spatial variability (Buytaert et al., 2006a) with a presence of the so-called "horizontal precipitation", which consist of fog and mist developed from the orographic uplift caused by the Andes (Buytaert et al., 2005b), which also

limits transpiration (Buytaert and Beven, 2011). Although, this additional source of water is minor and mostly intercepted by arbustive vegetation (*Chuquiraga*), other studies (Crockford and Richardson, 2000; Foot and Morgan, 2005) showed that the *páramo* ecosystem can catch low energy rain, drizzle and fog moisture on their leaves, which conduct over 50% of rainwater directly to the volcanic ash soils of Ecuadorian highlands (Janeau et al., 2015).





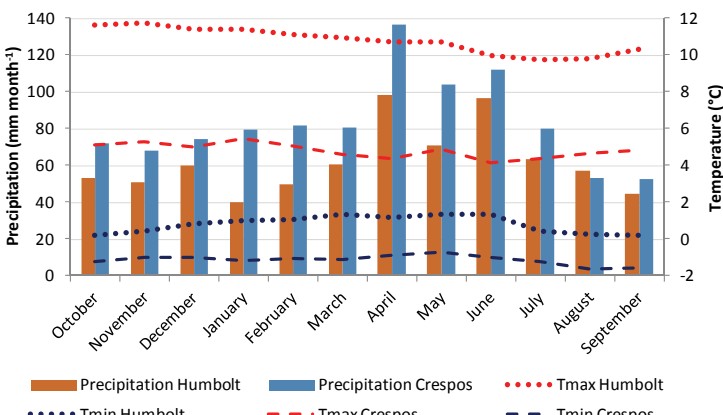

**Figure 2.** Average monthly precipitation, maximum and minimum temperatures at Humboldt and Crespos weather stations from 2000 to 2011.

## 3.    Data and methods

3.1  Data collection

The catchment is equipped with two hydro-meteorological stations (Figure 1) managed by the National Institute of Meteorology and Hydrology of Ecuador (INAMHI). The station located at the outlet of the catchment is the Humboldt station (4010 m a.s.l.), which records data of precipitation, water levels, temperature and electrical conductivity of the water. Isotopic and hydrochemical samples were collected in July 2014 during a 4-day sampling campaign. Figure 3 shows the catchment with

the sampling sites. The catchment contains two main tributaries that originate from the Antisana glacier. The principal difference is that the first one (subcatchment #2) flows through large boulders and rocks of different size until it meets the second one (subcatchment #9) that flows through *páramo* vegetation. The highest monthly average flow during the year occurs in June, with an average of 300 l s$^{-1}$; whereas the lowest monthly flow occurs in March with an average of 200 l s$^{-1}$. Due to the technical and logistics limitations in the study area, flow was measured only at the outlet of the catchment together with

precipitation values.

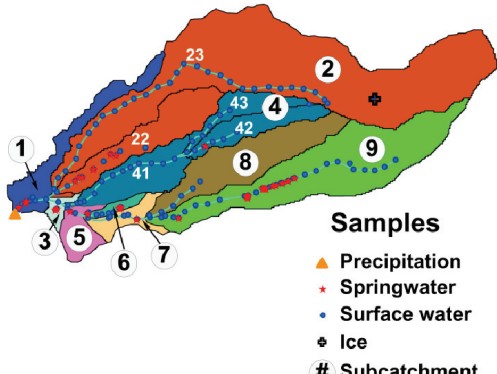

**Figure 3.** Subcatchment division with sampling locations for precipitation, spring water, surface water and ice.




### 3.2 Experimental set-up

#### 3.2.1 For dry conditions

In this work, dry conditions are defined as periods in which precipitation was absent for at least three consecutive days. Surface water samples were taken from the main streams including all tributaries ($n = 107$), spring ($n = 44$), ice ($n = 3$) as shown in Figure 3 during dry conditions. Every 200 m, electrical conductivity (EC) and temperature (°C) were measured *in-situ* using a WTW LF340 series conductivity meter. A volume of 2 ml was collected in a glass bottle (PTFE/silicone septa) filled to the top for stable isotopic analysis ($\delta^{18}O$, $\delta^2H$) to prevent evaporation. In addition, every 400 m two polyethylene bottles of 25 ml water samples were filtered for analysis of major anions ($Cl^-$, $SO_4^{2-}$) and cations ($Ca^{2+}$, $Mg^{2+}$, $Na^+$, $K^+$). The latter bottle was previously prepared for preservation by adding a drop of nitric acid ($HNO_3$). All vials were kept in a cooling box at 1 - 5°C.

Additional samples were also analyzed for $SiO_2$ using the 8185 method of Silicomolybdate using a Hach DR890 Portable Spectrophotometer and for $HCO_3^-$ using a Hach Digital Titrator. Additional information such as weather conditions, GPS coordinates and a description of the sampling site was also recorded.

#### 3.2.2 For wet conditions

During rainfall events, the sampling of the surface water and precipitation was undertaken only at the Humboldt Station for rainfall-runoff analyses. Samples were collected with a resolution of 15-20 min during events.

Rainfall samples were collected using a self-made device that consists of a funnel (diameter 140 mm) with a micro filter at the neck level joined to a drip chamber and to a 60 cm PVC tubing connected at the end to a rigid needle (3.8 cm, 28G) embedded in rubber cork of a polyethylene vacuum packed bottle (1L). One device was used per rainfall event, with the sample extracted immediately after or during the event following the procedure of the IAEA (IAEA/GNIP, 2014).

All samples were analyzed for major anions, cations and isotopes using the same procedure as described above in section 3.2.1 and $SiO_2$, EC and temperature was measured *in-situ*.

### 3.3 Laboratory Methods

Oxygen and hydrogen isotopic values, expressed in ‰ in relation to the Vienna Standard Mean Ocean Water (VSMOW), were measured using the Liquid-Water Isotope Analyzer (LGR DLT-100, precision <0.3‰ for $^{18}O/^{16}O$ and <1.0‰ for $^2H/^1H$).

The samples for major cations were determined by mass spectrometry using the Thermo Fisher Scientific XSeries 2 ICP-MS (limit of quantification ~2 ppb). Anions were analyzed by using ion chromatograph Dionex ICS-1000 (limit of quantification 2000 ppb). All analyses were performed following quality assurance and control procedures of the laboratory at UNESCO-IHE, Delft, The Netherlands.

### 3.4 Data analysis

#### 3.4.1 Spatial hydrochemical characterization

One-way ANOVA tested for significant differences ($P < 0.05$) for the different runoff sources e.g. ice, precipitation, surface water and spring water (shallow subsurface flow). The same test was applied for a location analysis related to subcatchments (Figure 3). If the results showed a large variation at subcatchment level, those were again divided in smaller subcatchments for a detailed analysis to identify the cause. Significant *t*-tests were followed by Tukey multiple comparisons as post-hoc tests indicated by a lowercase letters on top of each boxplot.

In addition, to assess the spatial distribution of hydrochemical parameters in the catchment, concentrations were plotted against distance from the outlet.





### 3.4.2     Flow pathways and routing

Tracers enabled the identification not only of the runoff sources but also the quantity they contribute to the river flow. We applied the mass-balance approach from downstream to upstream to calculate the discharge when two streams met at the confluence point (Eq. 1 and Eq.2)

$$Q_T = Q_1 + Q_2 \qquad [1]$$

$$C_T\,Q_T = C_1\,Q_1 + C_2\,Q_2 \qquad [2]$$

, where $Q_T$ is the total runoff, $Q_1$, $Q_2$ are the runoff components in m$^3$/s and $C_T$, $C_1$, and $C_2$ are the concentrations of total runoff, and of the runoff components in mg/l or ‰.

### 3.4.3     End Member Mixing Analysis (EMMA) and Hydrograph Separation

An End Member Mixing Analysis (EMMA) based on the method described by Christophersen & Hooper (1992) was carried out using the water quality parameters obtained. Mixing diagrams of EC (μS/cm), SiO$_2$ (mg/l), Cl$^-$ (in mg/l) and δ$^2$H and δ$^{18}$O (‰VSMOW) indicated their suitability as tracers for the hydrograph separation.

Isotope and hydrochemical data were combined with discharge data to perform three-component hydrograph separations based on steady state mass balance equations and hydrograph separation assumptions (Buttle, 1994; Pearce et al., 1986; Uhlenbrook

et al., 2002). A third runoff component was included in Equations 1and 2 to calculate three-component hydrograph separations for the total runoff ($Q_T$) (Eq. (3)).

$$C_T\,Q_T = C_1\,Q_1 + C_2\,Q_2 + C_3\,Q_3 \qquad [3]$$

Rainfall characteristics, including duration, total rain, maximum and average intensity were estimated for the rain events. A rainfall event was defined as a rainfall occurrence with rainfall intensity greater than 1 mm/hr, and intermittence less than four

hours. Peak flow, water depth, and time to peak were determined for each event.

Analytical and tracer end-member uncertainties were accounted for the hydrograph separation and quantification of the runoff components based on a Gaussian error propagation technique with 70% confidence interval (Eq.(4)) (Genereux, 1998).

$$W = \left\{ \left[ \frac{\partial y}{\partial x_1} W x_1 \right]^2 + \left[ \frac{\partial y}{\partial x_2} W x_2 \right]^2 + \cdots + \left[ \frac{\partial y}{\partial x_n} W x_s \right]^2 \right\}^{\frac{1}{2}} \qquad [4]$$

, where $W$ is the uncertainty of the each runoff component in %, $Wx_1 x_2$ are the standard deviations of each end-member, $Wx_s$

is the analytical uncertainty and $\frac{\partial y}{\partial x}$ are the uncertainties of the runoff component average contribution regarding the tracer concentrations.

## 4.     Results

### 4.1  Hydrochemical catchment characterization

#### 4.1.1     Runoff sources

All samples were clustered in four major groups based on the runoff sources: Ice ($n = 3$), Precipitation ($n = 4$), Surface water ($n = 107$) and Spring water ($n = 44$) and the groups compared for significant differences. SiO$_2$, K$^+$ and stable isotopes (δ$^2$H and δ$^{18}$O) gave a first glimpse of the composition of these groups by showing such differences amongst all groups (Figure 4). Major ions and EC values in Ice and Precipitation showed lower values in comparison to the other groups, which require a distinctive classification in order to give more informative results. For this reason, surface and spring water samples were

disaggregated and grouped per subcatchment.





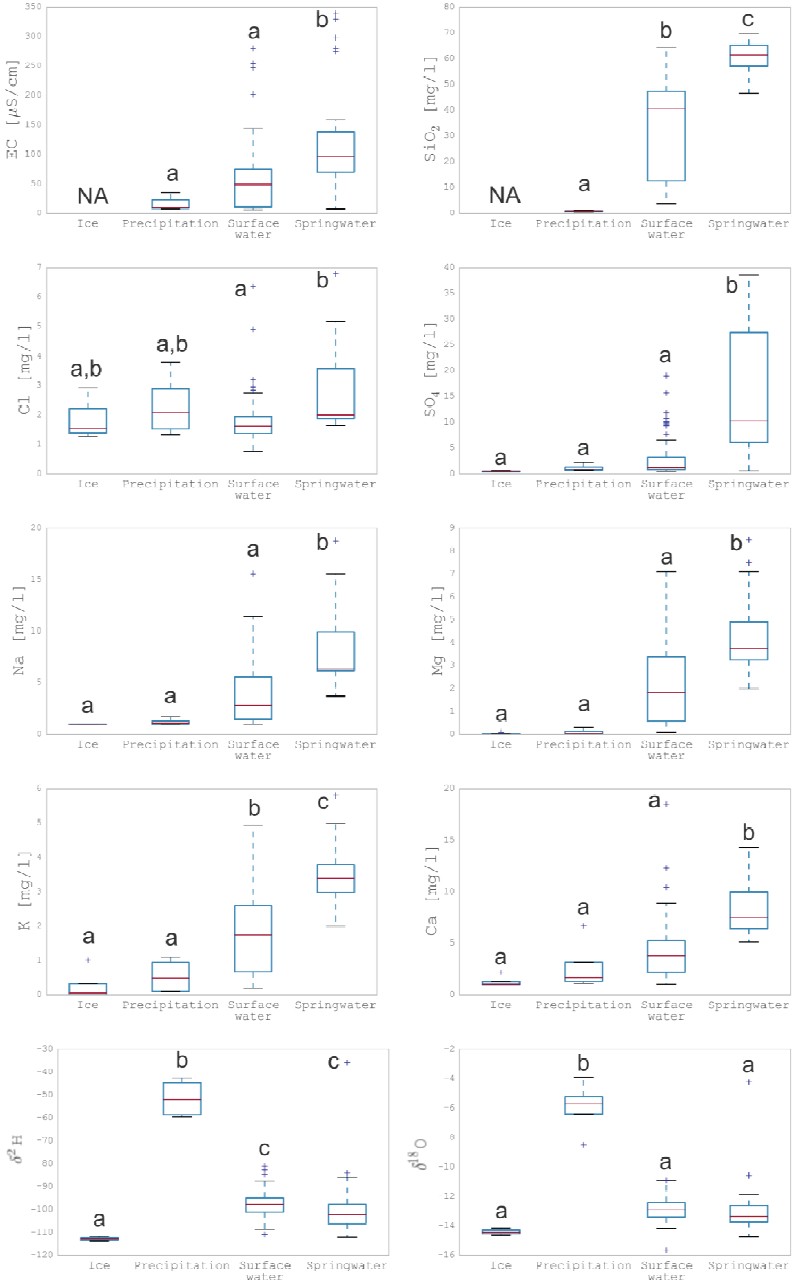

**Figure 4.** Chemical components and stable isotopes of water samples within the Los Crespos-Humboldt basin, analyzed per runoff source (Ice, Precipitation, Surface and Spring water). Lowercase letters indicate significant differences among sources ($P \leq 0.05$), according to Tukey's test. NA= samples are not available.





*Surface runoff – Subcatchment analysis*

In this second approach, only samples within the surface water category were classified in catchment and subcatchment groups to give more confidence intervals in the results (refer to Figure 3).

The major ions and EC values from the subcatchments 41, 42, 43, 6, 7, 8 and 9 exhibited no clear contrasting patterns among them, thus can be considered as a single group (Figure 5). These subcatchments belong to catchments 4 and 5 which lie in a highly vegetated side of the catchment. Whereas subcatchment 21, 22 and 23 that belong to catchment 2 were significantly different ($p \leq 0.05$) and therefore should remain grouped separately (Figure 5). Catchments 4 and 5 from now on will be referred to as *páramo* catchment. Catchment 2 lies on a mixed catchment of water coming directly from the glacier (subcatchment 23) without any other type of contribution and water from a small tributary of a combined source of surface and spring water (subcatchment 22). Subcatchment 23 from now on will be referred to as glacier catchment.

All major cations showed significantly lower values of concentrations for subcatchment 23, thus demonstrating a unique hydrochemical characteristic of streams derived from glacier components.







**Figure 5.** Major ions and EC values of water samples from surface water within the Los Crespos-Humboldt basin, analyzed per catchment and subcatchment. Lowercase letters indicate significant differences among catchments ($P \leq 0.05$), according to Tukey's test. NA= samples are not available.





*Surface runoff – Flow paths and routing*

Each of the previous analyses was important for the spatial hydrochemical characterization of surface and spring water within the Los Crespos-Humboldt catchment. A PCA was carried out to identify interrelationships between major ions, in which a smaller set was selected: EC, $SiO_2$, $Na^+$, $K^+$, $\delta^{18}O$, and $\delta^2H$ for further analysis.

5    Distance to the outlet and altitude were highly correlated ($P < 0.001$) and assumed to have a linear response despite the weather conditions which can differ at different locations within this high altitudinal mountain ecosystem. A distance to the outlet was selected to display the effect of surface water within the subcatchment and the confluence with other tributaries along the way to the outlet (Figure 6). The major ions and EC values showed a significant spatial variability and evidently separate two contrasting groups: surface water directly coming from the melting of the glacier (subcatchment 23) and the

10    surface water that comes from the vegetated areas (subcatchments 41, 42, 43, 5, 6, 7, 8, 9). It should be noted that subcatchment 9 starts also with a small contribution of glacier but most of the water comes from the *páramo* vegetated areas. The high concentration values of subcatchments 22 and 31 are due to a considerable amount of spring water contribution to the main channel as well as the small variations displayed among samples within the same subcatchments.





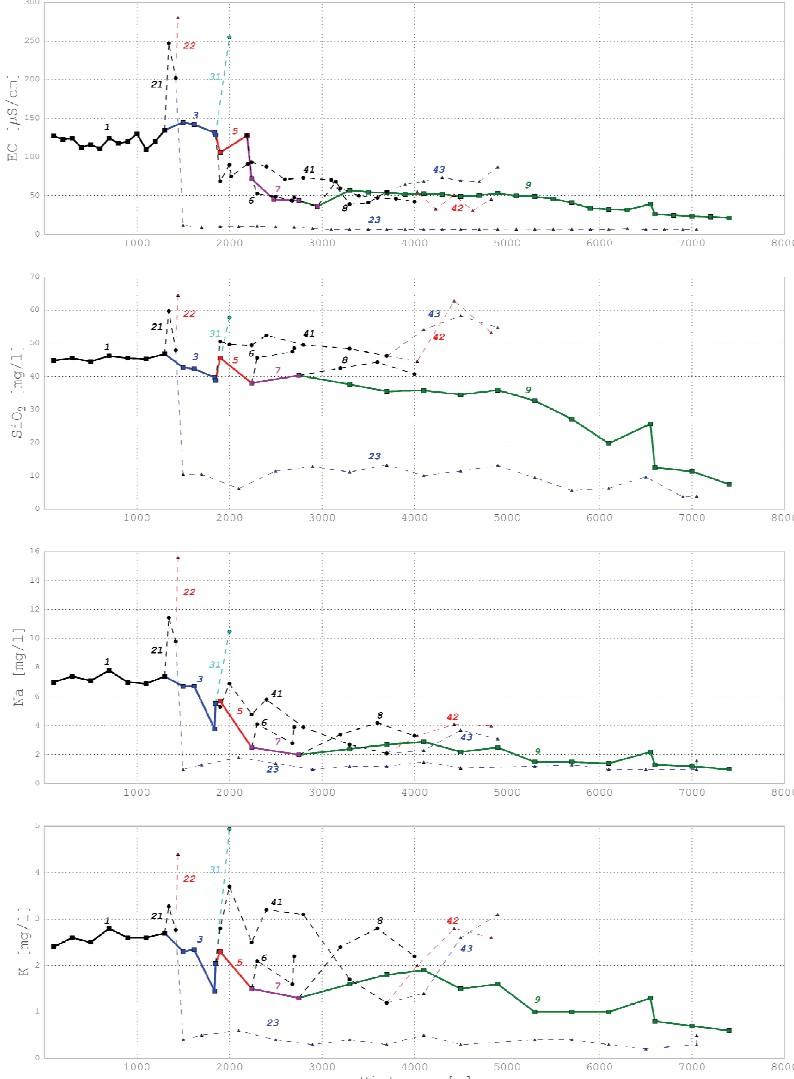

**Figure 6.** Chemical components of surface water samples within the Los Crespos-Humboldt basin, analyzed by distance to the outlet, numbers indicate subcatchment group. The thick solid line represents the main stream and dashed lines are the tributaries.





### 4.2  Water sources isotopic signatures

The sampling during rainfall events corresponds to low-medium intensity rain (2 mm total rain) and one of them was considered as representative and analyzed for rainfall-runoff evaluation. The duration of the event was 12 hours with a maximum intensity of 0.3 mm/h and an average intensity of 0.18 mm/h.

5   Isotope composition for all samples in the catchment is shown in Figure 7. Precipitation ranged from -8.5 to -3.9 ‰ for $\delta^{18}$O and from -59.5 to -42.4 ‰ for $\delta^2$H. Ice samples have a lighter isotopic signature, the lightest value corresponds to a 30 cm-depth sample, followed by a 20 cm-depth sample and the slightly enriched value belongs to the 10 cm-depth sample. The samples were clustered in such a way that the isotopic signature of the main river and tributaries that come from the *páramo* component were clearly identifiable, along with the glacier component. The signature of the *páramo* component shows a

10   relatively lighter or more depleted value of isotopes in comparison to the signature of the glacier component, which are heavier or enriched. Spring water signature showed a wide range of isotopic composition from -14.8 to -10.6 ‰ for $\delta^{18}$O and from -112 to -84.1 ‰ for $\delta^2$H. Likewise, the isotopic composition of storm runoff ranges from -13.03 to -11.2 ‰ for $\delta^{18}$O and from -99 to -89.3 ‰ for $\delta^2$H. The storm runoff samples are the ones collected in the main stream at the outlet of the catchment and include samples of pre- and post-event, which will be identified in the mixing plot analysis (Figure 8).

25




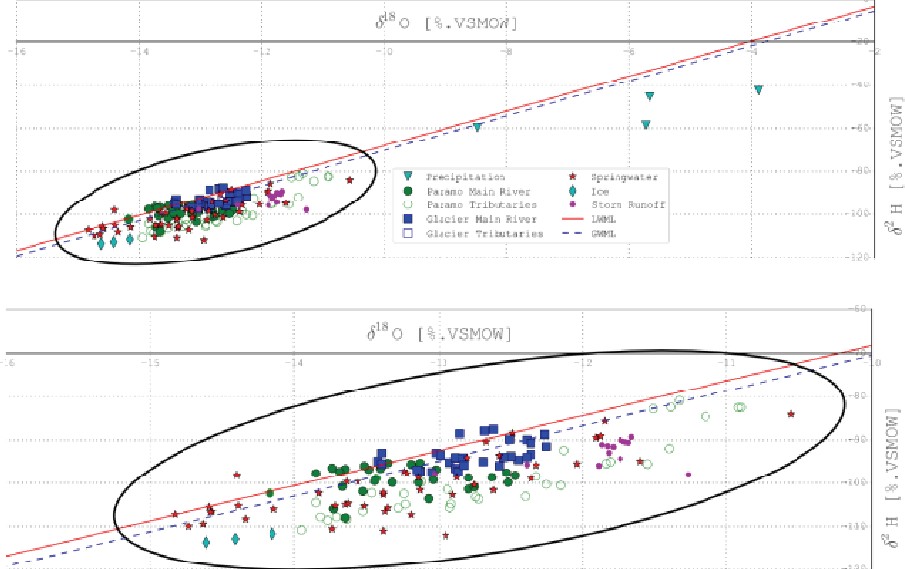

**Figure 7.** Stable isotope compositions of precipitation, surface water, spring, ice, storm runoff. GMWL: $\partial^2 H = 8.13 \times \partial^{18} O + 10.8$ ‰ (Source: Rozanski et al., 1993). LMWL for Izobamba: $\partial^2 H = 8.1 \times \partial^{18} O + 12.8$ ‰ (Source: IAEA, 2016)

### 4.2.1 Mixing plots

5    Mixing plots were derived with all possible permutations of the small set of parameters (EC, $SiO_2$, $Na^+$, $K^+$, $\delta^{18}O$, and $\delta^2H$). Three main components were considered: precipitation, glacier and *páramo*, each of which has its own chemical signature and serves as a vertex of a triangle that defines the boundaries of the storm runoff concentrations. EC and stable isotopes ($\delta^{18}O$, and $\delta^2H$) were identified as conservative tracers that characterize the end-member concentrations represented by precipitation, glacier and *páramo* runoff (Figure 8). Most concentration points of the storm water were located between the

10   runoff from *páramo* and glacier. The mixing plot shows the evolution of the stream water before (pre), during (event) and after (post) the event. The discharge started with high EC values of around 130 µs/cm, decreasing to an average of 70 µs/cm during the event and rising up to 95 µs/cm after 12 hours of the event. During the event the storm water is showing slightly heavier isotope values.





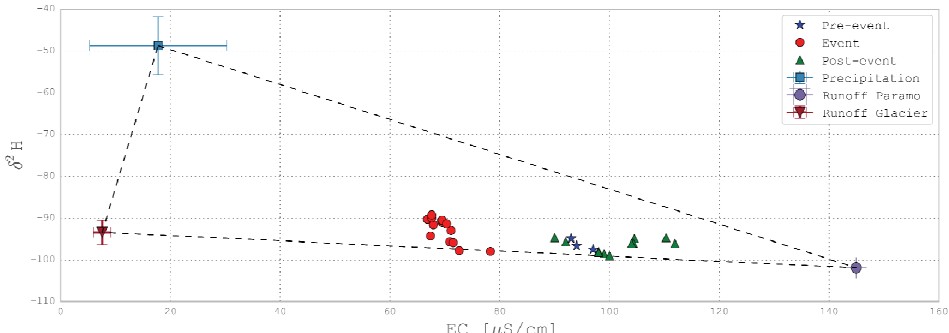

**Figure 8.** Mixing diagram showing stream water evolution and end-member EC and stable isotopes $\delta^2$H.

### 4.2.2 Hydrograph separation

Dry conditions

An initial estimation of the flow percentages during rainless periods was derived with EC, stable isotopes, and major ions

independently. The flow path estimations for major ions were not considered because the concentrations and/or differences were too low thus giving unrealistic estimations for the subcatchments. Thus, EC values and stable isotopes were used to estimate the contribution of water from the glacier component, which was of 21% for EC, 14% for $\delta^2$H and 15% for $\delta^{18}$O. Likewise, the percentages of flow that comes from the *páramo* vegetated areas are 52% for EC, 71% for $\delta^2$H and 78% for $\delta^{18}$O, the remaining percentages come from small streams that join the main stream close to the outlet.

Wet conditions

A three-component hydrograph separation based on EC and $\delta^2$H concentrations quantified the relative contribution of precipitation, glacier and *páramo* to the total flow. During the event, the total discharge was composed of 8% precipitation, 41% flow from glacier and 51% flow from the *páramo* component (Figure 9). The glacier component was the first to rise; the precipitation and *páramo* components have the maximum contribution during the peak time of the discharge. The rising

limb mainly comprised similar contributions from glacier and *páramo* components and to a lesser extent by precipitation, whilst during the recession limb the contribution of precipitation increased. After the storm runoff there was still contribution from the *páramo* component while the contribution from the glacier decreased gradually as shown in Figure 9.





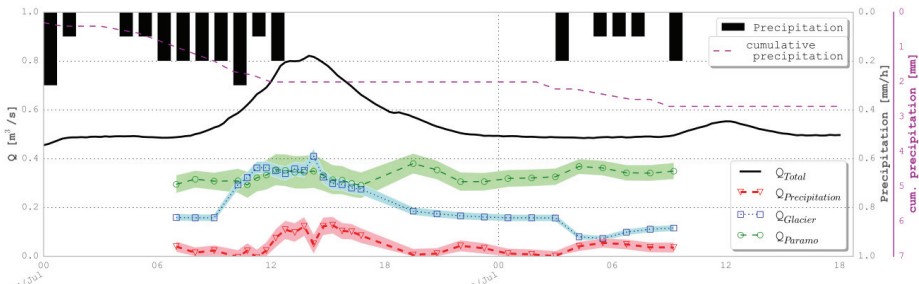

**Figure 9.** Three-component hydrograph separation, contribution of Glacier, *Páramo* and precipitation to stream runoff based on the EMMA using EC and $\delta^2H$ as tracers for 14 July rainstorm event. Colored area shows the estimated error propagation of the components.

## 5. Discussion

### 5.1 Spatial hydrochemical analysis and suitable tracers

Silica ($SiO_2$) and EC were proved to be appropriate tracers to separate the main groups of runoff sources (precipitation, surface and spring water); whereas all major ions succeed only to significantly distinguish the spring water from the rest of the sources. In all cases, the spring water displayed higher values with large variations; this suggests that some runoff could originate from deeper sources, from fissures and fractures in the rock, where longer residence and contact times have increased the ion concentrations. Similar evidence was observed in other headwater catchments (Hugenschmidt et al., 2014; Scanlon et al., 2001). However, attention should be paid to the outliers, which provide clear evidence of meltwater resurgence from the glacier. The latter confirms the meltwater infiltration during a study of the influence of the glacier on the water levels in the same catchment (Cauvy-Frauníe et al., 2013). Unfortunately, it is very difficult to crosscheck the water chemistry with the water signatures from the isotopes since they could be altered as a consequence of the strong influence of bedrock substrates, altitude and manifold underground processes (Nelson et al., 2011).

The analysis of surface water per subcatchment identified two distinctive groups, the samples from subcatchments 4 and 5 were pooled together and defined as the *páramo* component. Similarly samples from subcatchment 23 were defined as the Glacier component since that was the stream that exclusively carried water from the melting glacier. However, during dry conditions in the diurnal cycle the contribution of the glacier might: a) increase as a consequence of the melting of the glacier due to the strong shortwave radiation especially in the low part of the glacier, and b) decrease as consequence of sublimation of the glacier due to high wind velocities (Favier et al., 2004).

EC, silica and cations were appropriate tracers that displayed the significant difference between the glacier and the *páramo* component. The landscape played an important role influencing the hydrochemical composition of the surface water,





confirmed by the analysis of its distance to the outlet. The latter revealed the contribution of the spring water to the stream by showing changes in the concentration of the different tracers along each subcatchment and the shift in concentration while joining other subcatchments.

The stable isotopes ($\delta^{18}$O and $\delta^{2}$H) proved to be good tracers to distinguish between precipitation and ice samples; however

they were unable to clearly distinguish between surface and spring water sources. This could be partially related to the mixing of surface water samples that were taken along the streams encompassing a combination of surface and spring water. Large variations in the stable isotopes could also be associated with the undetermined quantification of shallow and deep subsurface flow.

### 5.2 Quantifying the contribution to storm runoff

Separation the contribution of each runoff component was challenging. The contributions may vary to an unknown amount due to unidentified layers and/or fissured and fractured rocks from which spring water originates. Usually, small head watersheds are a result of a mixing between rainfall, soil water and groundwater (Marechal et al., 2013). Nevertheless, since the water sampled in the rivers took into account the contribution of spring water, our main objective remains in the quantification of the relative contribution from glacier and *páramo* and their evolution with time as the main components for

the total runoff at the outlet of the catchment.

The isotopic composition of rainfall and their relative distance to the GWML propose a possible evaporation effect. Spring water signature has a wide range; those that showed heavier isotopic composition could be associated to precipitation, wetland or shallow subsurface recharge. Conversely, those with depleted compositions could be linked to deeper subsurface layers, and moreover, the high EC concentrations imply that the water could have been stored for a longer period of time.

During the event, the storm water either contained heavier isotopes that came from the rain, or could be associated with wetlands/open waters or shallow subsurface flow (from previous rainfall events), and also with contributions from the saturated zone, which can be highly dynamic. The challenge to separate the runoff components in this catchment was investigated in earlier studies (Mena, 2010) that estimated an average contribution of 45% of the glacier component, slightly above the estimation of 41% reported in the present study. Natural *páramo* can vary around 50 to 70%, while this study

reported around 51% (± 5% uncertainty). The remaining contribution from precipitation can be attributed to direct superficial runoff.

A representative end-member mixing analysis was carried out with three main components: precipitation, glacier and *páramo* as justified earlier. Some of the monitored stormflow samples were not fully confined within the triangles, which might increase the uncertainty in the evaluated event.

In order to overcome the limitations of non-conservative behavior of major ions, the application of stable isotopes for the hydrograph separation was preferred, a technique used commonly in tropical and subtropical areas (Elsenbeer, 2001; Goller et al., 2005; Klaus and McDonnell, 2013; Mul et al., 2008). In the present work, a plausible approach was the combination of EC and $\delta^{2}$H that demonstrated a clear separation of the three distinctive components: precipitation, glacier and *páramo*



likewise stated by Mena (2010). It is important to realize that these types of tropical Andean catchments havehigh spatial variability of precipitation (Buytaert and Beven, 2011; Buytaert et al., 2006b; Celleri et al., 2007) due to orographic effects, which to a certain extent can influence and might mislead the quantification of the contribution of this component.

Based on the hydrograph separation, the contribution of the glacier component during the storm increases at a faster rate. This is mainly attributed to the fact that there is no water retention at any point in that subcatchment and yet the riparian area consists mainly of boulders, rocks and large soil particles that drain rapidly with a very low water retention capability. Conversely, in the *páramo* region there are several zones with less steep slopes that are hydrologically disconnected due to the irregular terrain (Buytaert and Beven, 2011). Yet, they behave as floodplains, swamps and wetlands that dissipate the stream energy and buffer the peak flow at the outlet, contrary to what was found in a similar study by Buytaert *et al.* (2010). The soils of the riparian zone in the *páramo* subcatchments comprises smaller soil particles that are poor in percolation, thus offer a high water-holding capacity (Minaya et al., 2015) and consequently high water attenuation (Buytaert and Beven, 2011). Equally important is the interception of rainfall, which is the first process in a rainfall-runoff event. This interception particularly in the tussock vegetation should not be neglected and might contribute to a longer lag time during rainfall events and an increased recharge of water into the soil in these high altitude ecosystems (Buytaert and Beven, 2009; Janeau et al., 2015).

## 6.   Conclusions

The analysis estimated relative contributions of the main runoff components and provided valuable information on the origin of water and hydrologic characteristics and hydrochemical composition of the water cycle in this high mountainous region. This runoff ratio cannot be assumed to be maintained in the future because it is linked to future climate and land-use drivers. However, adequate water resource management should include the enhancement of the protection of the *páramos* as reservoirs of water in the highlands since they are the main contributors to runoff generation.

Rainfall events were monitored with medium intensity and the runoff patterns are in line with the expected dynamics within this catchment. The high contribution of the glacier component during rain events remains valid. Two sources with clear evidence of resurgence of meltwater from the glacier were identified, characterized consistently by low values of EC. It should be noted that during rainless times, there might be a higher variability due to the diurnal cycle and contribution of melt water due to the exposure to solar radiation. Therefore, the effect of temporal resolution needs to be further studied since these streams depend on the glacier influence. Certainly, long-term analysis will contribute to a better understanding of the dependency of runoff generation on soil moisture and vegetation interaction.

It is concluded that the present study is a spatial representation of the main runoff components. Further investigation is recommended with respect to the separation of shallow and deep subsurface flows. Moreover, the clustering of groundwater movement into a single group as 'spring water' offers further scope for investigation. The lack of soil moisture measurements and assumptions of the permanently saturated zones added uncertainty to the quantification of the subsurface processes that regulate the contribution of surface runoff. Despite these limitations and uncertainties, the combination of stable isotopes and





geochemical tracers improved the understanding of runoff processes in this combined glacier and *páramo* catchment in the Ecuadorian Andean region, for which no runoff investigations were available before.

## Acknowledgements

Financial support came from SENESCYT (Secretaría Nacional de Educación Superior, Ciencia, Tecnología e Innovación)

and from the Dutch Ministry of Foreign Affairs (DUPC program at UNESCO-IHE). Other institutions that also cooperated in the provision of key information and data are EPMAPS, EPN, INAMHI, INIGEMM. The authors express their sincere gratitude to Fred Kruis, Ferdi Battes and Berend Lolkema for their assistance in the laboratory analysis. The opinions expressed herein are those of the author(s) and do not necessarily reflect views of any of the Institutions named above. Thanks also to Aline Saraiva Okello, Lydia Cumiskey; and to Gareth Bird for proof-reading the manuscript.

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
