# Peer review of "Quantification of runoff generation from a combined glacier and *páramo* catchment within an Ecological Reserve in the Ecuadorian highlands"

_Hydrology and Earth System Sciences, 2016_

## Referee Comment (RC1) · Anonymous Referee #1 · 11 Jan 2017

The paper entitled "Quantification of runoff generation from a combined glacier and paramo catchment within an Ecological Reserve in the Ecuadorian highlands " by Minaya V., Suarez, V.C., Wenninger, J., Mynett, A., aims to identify the various runoff sources in a small mountainous glacierized Andean catchment using environmental tracers (stable isotopes and major ions) and hydrochemical features. This is a good piece of work that intends to answer a present and interesting question in the glacierized catchments in general. The approach is not novel in itself but for this precise catchment this is a new way to quantify the sources of the runoff sources. I think this article needs major revision in order to be published in HESS. Here you will find 4 major comments and some specific comments that should be taken into account for the next

submission.

GENERAL COMMENTS

GC1 : A first problem in this study is the characterization of the different sources of the runoff. Authors identify for instance in the figure 9 "Qprecipitation", "Qglacier", "Qparamo" but they identify also "Springwater" (Figure 7). The typology and the definition of each sources of water have to be clear. Furthermore "Qprecipitation" is not a good term. I understand that the definition is not easy but as the authors wrote, the hydrochemical signature has to be the driver to differentiate the sources. Other studies done in the same catchment prove that water originating from the glacier melting could give springwater, you should consider this point. Can we consider that "Qparamo" is a groundwater? Please respond to this in the article. A strong definition of groundwater has to be done. In order to be more accurate, a good description of the geology and the soil has to be conducted. In the part 4.1.1, a short description of the different samples is done, but for the springwaters more details are needed. How are the springs, in which kind of rocks? Are we sure that each sampling point correspond to one spring? GC2 : A additional table is needed with the details concerning all the samples (location, type of water, main characteristics, etc. . ..). GC3 : The weathering processes are not described in the paper even if some general statements are written. A good description of the geology and the different soils is needed. It would be interesting to give some quantification concerning the residence time of the water: (i) in the soil, (ii) in the fracturated aquifer. Page 16, authors stated that to increase the conductivity the water should be stored for long period of time in rocks but, in fact it depends on the type of the rocks, in evaporitic rocks for instance, the mineralization is very fast. GC4 : The EMMA methodology is briefly described in the section 3.4.3 but its application should be more detailed. Authors explain that they use only the runoff at the outlet, so I deduced that this runoff is QT. In the figure 9 authors stated that the EMMA analysis is done with 2 variables, EC and $\delta$2H. How can we calculate the 3 different terms if one considers all the other unknown factors? What are the different equations composing

the system and how authors solve this equation system?

SPECIFIC COMMENTS Title – The title should be more informative, what is the study period? What methodology is applied ? Which period is studied? Abstract: The altitudinal range of the catchment should be specified. The area of the glaciers should be specified too. Abstract P.1, l.14 - The term "Andean region" is inappropriate as the study is focused only on one study case. Not all the catchments in the Andean region are volcanic. Please be more specific. Abstract P.1, l. 20 – The "dry and wet conditions" have to be defined, it will depend of the considered timescale. If one see the figure 2 at monthly time step, we don't observe any dry season. P2., l.16 – Write "stable isotopes" instead of "isotopes" P5., l.4 – How be sure that 3 samples are sufficient to characterize the chemical signature of the ice? For me the number of analysis has to be increased, the number of samples is not sufficient enough. Considering basic statistics it is not possible to draw box-plots with only 3 points as samples! P6., l.30 – Once again , 3 samples for Ice and 4 samples for the precipitation are not sufficient to define a strong signal for these water types. P6., l.31: The number of spring water is n=44, could you precise if each point is an individual spring? For that the adding of a new table is necessary (see GC2). P12., l.5-6: : I don't understand why three (of the four) samples of precipitation water are very far from the GMWL and LMWL curves. I suspect some problem due to evaporation during the sampling and/or during the storage. Please explain why we observe these differences. Why the 3 samples of ice are not located on the LMWL? Please define the acronyms GMWL and LMWL. P14., l.6-9: Please precise the period for the calculations of the different contributions. P14., l.11-17: Idem P15. l.11-12: The two references cited are not relevant because the type of catchment are very different. Please provide other references with catchments that have the same behavior than the catchment of your study (high catchment in volcanic tropical region). P16. L.23 and P17, l.1: The reference Mena (2010) is not freely available, has no DOI and so it should not be cited. Please provide other references. A table is missing with the indication for all the water samples, locations, main characteristics, etc. . ... Figure 1: The latitude and the longitude have to be added, the sources of data for each map

have to be mentioned. Figure 2: What is the time step for the temperature data? Monthly? If it is the case please indicate "monthly temperature". If the temperatures are monthly temperatures, it would be better to plot the values with points without a line between them. Figure 3: How is made the separation between the sub-catchment, is it topographic? Which is the DEM (source, resolution) and which methodology has been used to separate each sub-catchment? How to be sure that the springwaters are not superficial rivers (see GC 1)? Some sub-catchments are missing, for example 21, 31, 32. Please add these sub catchments to the figure. Figure 4: The font size is too small. How can you draw box plots for the categories "ice" and "precipitation" with only 3 and 4 values respectively? Concerning the $\delta$18O and $\delta$2H rates, how do you explain the differences between the two categories "Ice" and "Precipitation"? Figure 5: The font size is too small. It should be indicated that the numbers for the X-axis represent the number of the sub-catchment. How many samples are used to draw the different box-plots? The number n of samples has to be specify (may be in the new table). Figure 6: The font size is too small. Figure 7: see comment above (P12l.5-6). Figure 8: Please be more precise and define the three following terms: "pre-event" "event" and "post-event" Figure 9: No mention is done to Qspringwater : why?

---

## Referee Comment (RC2) · Anonymous Referee #2 · 18 Jan 2017

The submission deals with runoff generation and water sources in a high-mountain catchments dominated by a glacier and characteristic Andean grasslands (paramo). The work reports sampling of hydro chemicals and stable water isotopes along the stream and for one event at the catchment outlet. This data is supported by precipitation, ice, and spring sampling. The approach is not novel, yet the landscape and the crucial role of grasslands makes this study of interest for the scientific community, especially as high-mountain and glaciated catchments are under increased and constant stress. Yet, I wished that the senior authors would have invested more time in their student's submission rather than passing the work on to the reviewers. I feel that the paper needs quite substantial revisions before it can be accepted. I think that this work

can be a substantial contribution to the hydrological community after careful revision.

Major concerns:

i) What I am most critical about is the lack of a clear story line; especially the result section offers a bunch of data that were presented rather unstructured and it was hard to tease out the important bits of it. Why is what kind of data presented? I often felt lost, maybe also due to the lack of a proper description of the sampling.

ii) I felt confused throughout the read. There was plenty of sampling, along the stream, for events etc. What is clearly needed is a table and a more detailed description of the sampling.

iii) I felt that the use of the sampling along the stream was never really detailed used and the core of the works ends up as one hydrograph separation (which lacks the proper methodological description). What precipitation was used as end-member etc.?

iv) I am somewhat critical about the lack of precipitation samples, especially since they show quite some spread and plot below the LMWL. How does the station where the LMWL was recorded compare to the local conditions (such as elevation etc.). This could lead to a clear offset of the true local LMWL. There is no clear description where ice was sampled. Three samples is somewhat small, but at least the spread is small too.

v) The landscape units are poorly described in the manuscript, e.g. Figure 1 does not even present the paramos.

vi) The figures quality in general. Figures 1-3 are acceptable quality, figures 4-9 are not.

Minor concerns: The English could use a revision. The sentences are often nested and overly long. The title is not precise and too long. Please check the proper use of altitude, I think the differentiation between altitude and elevation needs some attention, cf. McVicar and Körner (2013), Oecologia. "On the use of elevation, altitude, and

height in the ecological and climatological literature" The abstract needs some rework. Paramo is never explained. L14-16 lacks precision. The results need to be more detailed.

P1L33-34: "...river's....resurgence" this is unclear. P1L39. The concern.... Check the grammar. P2L10ff: Please make the research gap clearer, really try to make an effort why exactly this study is needed. P2L15ff: Restructure and specify the research question (or use hypotheses). P2L33: "Location". For me this seems not right, as there is much more described here, such as catchment properties. P2L34ff. It was not always clear to me if the individual information was for the catchment, or the region. Furthermore, I found it difficult to follow since the authors jumped around in there references to distinct elevation ranges. What is low, what is high, etc. Please consider restructuring the section 2. P2L45-47: Not sure if some information that is clearly not linked to the catchment is needed. P4L10-11. Make it easier. Subcatchment#2 flows through.... and subcatchment#9... P5L4: Please introduce a table where details are described. P5L5: Every 200m, you leave it for the reader to guess. You should clearly state that this was sampling along the flowing stream channel. P5L10: reference for the method P5L14: You mean catchment outlet? How many events? P5L16: When were the precip samples taken? P5L29ff: The text would profit when transformed to active style. The PCA was not mentioned in the methods. P6L1: The header is unclear, I am also not sure what exactly was done with this mixing analysis. I stay confused. P6L29: Maybe "end member" would be a better term? P6L30ff.: See major concerns. Please avoid methods in the results, describe everything in section 3. P7: Figure 4: Box plots with 3 and 4 data points are sketchy. The quality of the figure needs to be improved, line width, font size, a, b etc is not mentioned in the captions P8L2: Avoid introductory sentences in the results paragraph. Straight to the point. What is important? P8L7-8, this should be mentioned in the study site. P9: Please reconsider the presentation in this figure. You leave it to the reader what might be of importance. Furthermore, the figure is not understandable just based on the captions. Font size, line width etc. should be improved. P11: completely revise P12L2-3: You refer to event and events.

How many? P12L2: 2mm rain is not an intensity. Was it just one event that had 2mm? These lines are confusing, I cannot figure out what was done and sampled (see suggestion about tables earlier). P12L2ff. You do not mention the LMWL here. How does the LMWL relate to the local conditions? How many km away? On what elevation? Etc. P13: The mixing plot and hydrograph separation section should be merged. P14: It is not clear here what end-members are used, but a merger with the mixing diagram section will help. Also consider presenting this information in the methods (which may not work, in case the mixing diagram was used to determine the end members, so just results in that case). P16L9ff. The discussion here needs some more work. You need to really make the point how your results improved both, the understanding of runoff generation in the catchment beyond previous understanding, and how this makes the work relevant for the same landscape at other places, and how the results compare to other researcher's work. The latter is needed to show the importance of the results for the community, you can close the story that you opened at the end of the introduction, where you should state (earlier comment) why this work is needed.

---

## Author Comment (AC1) · 14 Apr 2017

**Detailed point-by-point list of answers to the reviewers´ comments:**

**Anonymus Referee #1:**

> The paper entitled "Quantification of runoff generation from a combined glacier and paramo catchment within an Ecological Reserve in the Ecuadorian highlands " by Minaya V., Suarez, V.C., Wenninger, J., Mynett, A., aims to identify the various runoff sources in a small mountainous glacierized Andean catchment using environmental tracers (stable isotopes and major ions) and hydrochemical features. This is a good piece of work that intends to answer a present and interesting question in the glacierized catchments in general. The approach is not novel in itself but for this precise catchment this is a new way to quantify the sources of the runoff sources. I think this article needs major revision in order to be published in HESS. Here you will find 4 major comments and some specific comments that should be taken into account for the next submission.

We thank the referee for the valuable comments. Hereby we present a point-by-point reply to the referee´s questions and comments.

**GENERAL COMMENTS**

1) **GC1 : A first problem in this study is the characterization of the different sources of the runoff. Authors identify for instance in the figure 9 "Qprecipitation", "Qglacier", "Qparamo" but they identify also "Springwater" (Figure 7). The typology and the definition of each sources of water have to be clear.**

   The first approach during the analysis was a characterization of the main runoff sources being surface, spring, precipitation and ice as indicated in Section 3.2.1. However, based on the large variation of chemical components and stable isotopes of surface water and springwater, a separate analysis was performed. Surface water samples were analyzed related to catchment and subcatchments and springwater samples were analyzed per geological background. The latter was not included in the manuscript but based on the comments from the referees, it will be included in this improved version. The results from the surface water samples showed two distinctive groups: surface water coming from the melting of the glacier (subcatchment 23) and surface water that comes from the páramo-vegetated areas (subcatchments 41,42,43,5,6,7,8,9) as stated on pg 10 lines 8-14. In this regard, only the results from the surface water were taken into account further for the hydrograph separation analysis; since one of our objectives was to quantify the water coming from the two sources being páramo and glacier during dry and wet conditions.

2) **Furthermore "Qprecipitation" is not a good term. I understand that the definition is not easy but as the authors wrote, the hydrochemical signature has to be the driver to differentiate the sources. Other studies done in the same catchment prove that water originating from the glacier melting could give springwater, you should consider this point. Can we consider**

**that "Qparamo" is a groundwater? Please respond to this in the article. A strong definition of groundwater has to be done.**

Thanks for your comment, we have changed Qprecipitation for Qevent, which reflects exactly a distinctive component within the hydrograph separation during a rainfall event. Regarding the water originating from the glacier melting, in the Discussion section pg 15 lines 12-14, we discussed that the results from some outliers in the springwater group are a clear evidence of the meltwater resurgence from the glacier as also confirmed in other studies like Cauvy-Fraunié et al (2013). The large variations and high values of the chemical components and stable isotopes of the springwater samples suggest that some runoff could originate from deeper sources, from fissures and fractures in the bedrock. However, as stated in pg 15 lines 14-16: "*Unfortunately, it is very difficult to crosscheck the water chemistry with the water signatures from the isotopes since they could be altered as a consequence of the strong influence of bedrock substrates, altitude and manifold underground processes (Nelson et al., 2011)*". We cannot state that Qparamo (now Qevent) is entirely groundwater since it represents a mix of surface water, shallow subsurface flow and at some extent even deeper groundwater flow that comes to the surface via fissures and fractures of the bedrock. In this regard, a specific and strong analysis of groundwater should be done as further research and to complement this current study. We have added a statement in Section 5.2 to state what Qparamo (now Qevent) represents: "*The contribution from the paramo is a result from the mixing between surface runoff, soil water, shallow surface flow and groundwater as evidenced in other small head watersheds (Marechal et al., 2013)*".

3) **In order to be more accurate, a good description of the geology and the soil has to be conducted. In the part 4.1.1, a short description of the different samples is done, but for the springwaters more details are needed. How are the springs, in which kind of rocks? Are we sure that each sampling point correspond to one spring?**
The authors intention was to make a simplification of the work done by focusing in the quantification and further analysis of the water coming from the glacier and paramo subcatchments. However, we agreed with the referee in the sense that the description of the geology together with the analysis of the springwater will clarify some of the results that might be incomplete. We have added in Section 2:

"*Geology*
*The geology of the catchment has a wide detritic range that holds a variety of volcanic deposits from previous eruptions (Figure 1d), the last significant eruption occurred nearly 1000 years ago based on stratigraphic studies (Hall et al., 2012). The peak is slightly flat; it presumes that the crater is glacier filling. Although there is no volcanic activity or hot fumeroles lately, there are reports of $SO_2$ gas in higher elevations (Hall et al., 2012). Most of the stratigraphy is composed by dark layers of ash and andesite scoria, which is product of the fall of eruptive clouds with intercalations of fluvial deposits (Hall et al., 2012).*

*The geology as shown in figure 1d is composed, next to the glacier cover, of morraines, glacial-fluvial sediments, tillites, volcanic rocks and Lahar rojo. The morraines are deposited debris that form along the glacier from the receding of the glacier. These areas are characterized by lagune formations which intercept meltwater. The pleistocene lavas formation are older volcanic pyroclastic deposits which are composed of andesite rocks containing plagioclase, amphibole and feldspar minerals. The Hialina Lava is formed also of andesite content. However, this is a younger formation with olivine, plagioclase and quartz, arranged in a matrix formed by volcanic glass (Alvarado, 2009). The Lahar Rojo is a sequence of red volcanic lava deposits along the Antisana river. Its pyroclastic material when mixed with water became red indicating several volcanic eruptions during the Holocene.*

[Figure]

*Figure 1d Geology of the catchment the Los Crespos - Humboldt, Antisana icecap - Ecuador (original source from Hall et al (2012))*

In addition we have added in Section 4.1:

"*Spring water*

*Spring water characteristics were based on the geological formation from which they originate (refer to Figure 1d). Spring water samples that come from the Lahar Rojo formation were significantly higher in most of the major ions and EC concentrations (Figure 7). The rest of the geological formations showed different concentration ranges; however, they could not be tested for significance due to the lack of samples for specific major cations.*

[Figure]

*Figure 7 Chemical components and stable isotopes of spring water samples within the Los Crespos-Humboldt basin, analyzed per geological background (Ch = Chacana volcanic rocks, Hi = Hialina lava, LaRo = Lahar Rojo, LaPl = Lavas Pleistocene, Ti = Tillite late ice age). Lowercase letters indicate significant differences among geological background (P ≤ 0.05), according to Tukey's test."*

In Section 5.1:

*"In a more comprehensive analysis, most of the spring water samples showed silica concentrations of 55 to 70 mg/l; while the samples that correspond to the type of geology Tillita showed concentrations between 45 and 50 mg/l. In most of the cases the latter type of geology consists of impermeable tough layers that have a shallow water table (Cuesta et al., 2013) and thus could easily get in contact with the subsurface flow and experience a dilution effect. The comparatively higher concentrations in the cations ($Ca^{2+}$, $Mg^{2+}$, $Na^+$, $K^+$) in spring water showed similar characteristics to silica. Hence these cations should be considered as indicators for water from deeper soil layers in study areas comparable to this one. These results strengthen the assumption that most of the spring water comes from a groundwater source; nevertheless the discrimination among groundwater, shallow and deeper subsurface flows are subjects of further analysis which lies outside the scope of this study."*

In references:

Alvarado, C.: Caracterización hidrogeológica de ls vertientes occidentales del volcán Antisana como parte de los estudios de los glaciares y páramos frente al cambio climático (Unpublished dissertation), Thesis. Escuela de Ingeniería en Geología, Universidad Central del Ecuador, Quito, 2009.

4) **GC2 : A additional table is needed with the details concerning all the samples (location, type of water, main characteristics, etc. . ..).**

Thanks for the suggestion. We have added a table in the Annex section with all the details of the samples collected during our field campaign. Annex is included at the end of this response letter.

5) **GC3 : The weathering processes are not described in the paper even if some general statements are written. A good description of the geology and the different soils is needed. It would be interesting to give some quantification concerning the residence time of the water: (i) in the soil, (ii) in the fracturated aquifer. Page 16, authors stated that to increase the conductivity the water should be stored for long period of time in rocks but, in fact it depends on the type of the rocks, in evaporitic rocks for instance, the mineralization is very fast**
We have included a description of the geology in section 2 (also please see reply of comment #3). A brief description of the soils is given in pg 2 lines 44-52. In the discussion, we have included the following section regarding the weathering processes:

*"Catchment geology and weathering processes*

*The chemical signature observed in the surface water samples is in accordance with the geology of the area. Surface water samples obtained near the glacier and morraines formations (upper section of catchment 2) show low electrical conductivities (mean of 7.5 µS/cm) and silica concentrations (mean 9 mg/l), as well as other ions (Figure 5). Catchment 1, where the Hialiana lava formation is dominant, shows higher electrical conductivities (mean:*

*149 μS/cm) and silica concentrations (mean: 48 mg/l). The Hialiana lava formation is dominant in catchment 1. These area is rich in olivine, plagioclase and quartz. Although quartz are highly resistant to weathering processes, olivines are known for decomposing faster (Goldich, 1938; Appelo & Postma, 2005). The weathering of these minerals results in increased contents of silica, bicarbonates, and cations in the water. Sodium is also derived from weathering of plagioclase materials. Catchment 1 contains the highest sodium concentrations (mean: 7.5 mg/l) as shown in Figure 5. The Pleistocene lavas are dominant in catchment 4. These are characteristic for also their high silica content (mean: 52 mg/l), but as opposed to the Hialina lava, they contain lower sodium concentrations (mean: 4 mg/l). Magnesium is evidence of weathering of pyroxenes and amphiboles. Catchment 4 displays a wider range of magnesium concentrations from 1.4 to 5.7 mg/l. For both catchments 1 and 4, calcium is known to be released with the weathering of amphiboles and pyroxenes. These weathering processes may also result in the precipitation of carbonates and clay minerals. Surface water samples from the Lahar Rojo formation did not show the high ionic content expected as observed in the groundwater samples. Figure 7 (in this authors' response) shows the high concentrations observed in the Lahar Rojo section. The high clay content in this formation explains the high observed ionic content found in the groundwater samples. In catchments 5,6,7,8,9, the dominant formations are glacial fluvial sediments, tillites, lava and breccia. These sediment deposits have lower electrical conductivities thus lower ionic content, but a wide range in silica concentrations. Their low ionic content is explained by the source of these materials which comes from the glacial debris."*

In References:

Appelo, C. A. J. and Postma, D.: Silicate Weathering. Geochemistry, Groundwater and Pollution, Second Edition, pp. 375-414, Taylor & Francis, 2005.
Goldich, S. S.: A Study in Rock-Weathering The Journal of Geology, 46, 17-58, 1938.

No site-specific information/studies were found regarding the residence time of the water in the soil or in the fractured aquifer.

6) **GC4 : The EMMA methodology is briefly described in the section 3.4.3 but its application should be more detailed. Authors explain that they use only the runoff at the outlet, so I deduced that this runoff is QT. In the figure 9 authors stated that the EMMA analysis is done with 2 variables, EC and δ2H. How can we calculate the 3 different terms if one considers all the other unknown factors? What are the different equations composing C2 the system and how authors solve this equation system?**

*Indeed, we used only the runoff at the outlet. We have updated Line 13 on page 6: "Isotope and hydrochemical data were combined with discharge data taken at the outlet of the basin to perform three-component hydrograph separations based on steady state mass balance equations......"*

For the End Member Mixing Analysis, mixing diagrams with combinations of EC ($\mu$S/cm), SiO$_2$ (mg/L), Cl (mg/L), SO$_4$ (mg/L), Na (mg/L), Mg (mg/L), K (mg/L), Ca (mg/L), $\delta^2$H (‰ VSMOW), $\delta^{18}$O (‰ VSMOW) were created first. In addition, these parameters were plotted against discharge to observe the dilution bevaviour and hysteresis. Then a principal component analysis was carried out on the mentioned parameters indicating that two principal components explained 90% of the data variability leading to a three component hydrograph separation.

**SPECIFIC COMMENTS**

7) **Title – The title should be more informative, what is the study period? What methodology is applied ? Which period is studied?**
   We have added some informative details in the title that reflect the work done: "*Quantification of runoff generation from a combined glacier and paramo catchment during dry and wet conditions using environmental isotopes within the Antisana Ecological Reserve in the Ecuadorian highlands*"

8) **Abstract: The altitudinal range of the catchment should be specified. The area of the glaciers should be specified too. Abstract P.1, l.14 - The term "Andean region" is inappropriate as the study is focused only on one study case. Not all the catchments in the Andean region are volcanic. Please be more specific. Abstract P.1, l. 20 – The "dry and wet conditions" have to be defined, it will depend of the considered timescale.**
   Thanks for your comment, we have included the catchment elevation range in the Abstract section line 17: "*This study focuses on data collection and experimental investigations in a small catchment (15.2 km$^2$) that ranges between 4000 to 5700 m a.s.l. within the Antisana Ecological Reserve in the Ecuadorian Andean Region. It consist of 2.3 km$^2$ glaciers, 10.3 km$^2$ páramo grasslands and 2.6 km$^2$ moraines.*"
   Pg1 line 14 has been updated: "*However, not all runoff processes are well-understood in the Andean Region due to the high spatial variability of precipitation. Particularly in the northern Ecuador, young volcanic ash soil properties, soil moisture dynamics and other local factors such as vegetation interception and high radiation might influence the hydrological behaviour.*"
   Pg1 line 20 has been updated: "*.... in order to determine their respective contribution during dry and wet conditions. Dry conditions defined as periods in which precipitation was absent for at least three consecutive days and wet conditions during rainfall events.*"

9) **If one see the figure 2 at monthly time step, we don't observe any dry season.**
   We agree that the Figure 2 does not show a clear and defined dry season. The catchment is characterized for having precipitation throughout the year therefore the Figure shows period of less precipitation. As indicated in pg 3 line 11-12: There are two main sources of precipitation: those influenced by the air masses from the Amazon region and those influenced by the inter-Andean valley regime (Vuille et al., 2000).

10) **P2., l.16 – Write "stable isotopes" instead of "isotopes"**

Pg2 line 16 updated.

11) **P5., l.4 – How be sure that 3 samples are sufficient to characterize the chemical signature of the ice? For me the number of analysis has to be increased, the number of samples is not sufficient enough.**

Thanks for your comment. The amount of data is a limitation to make a complete analysis and chemical signature characterization of ice. We agree that we cannot draw strong conclusions on the chemical signature of ice based on a very small dataset. In this regard, the results were only used to have a first overview of the chemical components and stable isotopes comparing four different runoff sources being ice, precipitation, surface water and springwater. In a later stage we analyzed further only the surface water and springwater samples that have a larger number of samples.

12) **Considering basic statistics it is not possible to draw box-plots with only 3 points as samples! P6., l.30 – Once again, 3 samples for Ice and 4 samples for the precipitation are not sufficient to define a strong signal for these water types.**

As explained in the previous comment. We are aware of the limitation of our sampling points for ice, hence it was not used for further analysis on the quantification of the runoff coming from *páramo* and glacier subcatchments. In case of precipitation, the four samples correspond to the same rainfall event that was chosen for the End-member analysis and hydrograph separation. The amount of samples are limited to the amount of rainfall during the event.

We have updated Figure 4 and replaced the box-plots for other type of plot showing only the distribution of samples and their mean.

[Figure]

*Figure 4. Chemical components and stable isotopes of water samples within the Los Crespos-Humboldt basin of different runoff sources (Ice, Precipitation, Surface and Spring water). The blue dots represent the samples and the red star the mean values. NA= samples are not available.*

**13) P6., l.31: The number of spring water is n=44, could you precise if each point is an individual spring? For that the adding of a new table is necessary (see GC2).**

Indeed the number of samples correspond to an individual spring which are detailed in the Annex section. Please check table at the end of this letter.

**14) P12., l.5-6: : I don't understand why three (of the four) samples of precipitation water are very far from the GMWL and LMWL curves. I suspect some problem due to evaporation during the sampling and/or during the storage. Please explain why we observe these differences.**

During the analysis, these precipitation samples for stable isotopes also called our attention since they are deviated from the LMWL and GMWL slopes. It's unlikely that the main reason is due to evaporation during sampling and storing since we followed strictly the procedure of the IAEA as indicated in pg5 lines 18-19. Therefore, we hypothesize that the distance of the precipitation samples to the LMWL and GMWL lines might indicate other secondary evaporation processes that occur when raindrops fall in a warm atmosphere.

These is suggested by the values of deuterium excess less than 10‰ in all precipitation samples. The event was not a heavy rain and therefore the raindrops are slightly more enriched.

We have included a small statement about our possible explanation. However, an in-depth and further analysis are not within the scope of this study. It will definitely need a robust number of samples to draw strong conclusions. Pg 16 Line 16: *"The isotopic composition of rainfall and their relative distance to the GWML propose a possible evaporation effect. The first raindrops are usually more isotopically enriched (Gat & Carmi, 1970). For the specific case of precipitation, further research on rainfall events at this location should be done to check for possible re-evaporation processes and contributions of different water vapor sources that might occur taking into account inter and intra event variability in the hydrological process."*
In References:

Gat, J. R. and Carmi, I.: Evolution of the isotopic composition of atmospheric waters in the Mediterranean Sea area, Journal of Geophys. Res., 96, 13179-13188, 1970.

**15) Why the 3 samples of ice are not located on the LMWL? Please define the acronyms GMWL and LMWL.**

The relative distance of the samples of ice to the LMWL and GMWL curves are most likely due to evaporative losses during the melting. The vials available for sampling were not adequate for ice. They were completely full during sampling but unfortunately the time between the sampling and the storage was long and led to evaporation processes and isotopic fractionation during melting as seen by the excess of headspace within the vial.

Regarding the acronyms, thanks for the observation. We have defined the acronyms that were missed in Figure 7, as follows:

*"Figure 7. Stable isotope compositions of precipitation, surface water, spring, ice, storm runoff. Global Meteoric Water Line (GMWL): $\partial^2 H = 8.13 \times \partial^{18}O + 10.8$ ‰ (Source: Rozanski et al.,*

*1993). Local Mean Water Line (LMWL) for Izobamba: $\partial^2 H = 8.1 \times \partial^{18}O + 12.8\,‰$ (Source: IAEA, 2016)"*

**16) P14., l.6-9: Please precise the period for the calculations of the different contributions.**

We have updated the statement with the sampling period on Pg 14 line6: *"Thus, EC values and stable isotopes were used to estimate the contribution of water from the glacier component, which was of 21% for EC, 14% for $\delta^2 H$ and 15% for $\delta^{18}O$ during the sampling campaign on July 4 - 7, 2017."*

In addition, we have added this information in Section 3.1 so it is clear that the period is different for dry and wet conditions, pg4 line 8: *"Isotopic and hydrochemical samples were collected in a sampling campaign carried out in July 2014. For dry conditions July 4-7 and for wet conditions July 14-15."*

**17) P14., l.11-17: Idem P15. l.11-12: The two references cited are not relevant because the type of catchment are very different. Please provide other references with catchments that have the same behavior than the catchment of your study (high catchment in volcanic tropical region).**

Thanks for the observation. We have deleted the statement to avoid a misunderstanding.

**18) P16. L.23 and P17, l.1: The reference Mena (2010) is not freely available, has no DOI and so it should not be cited. Please provide other references.**

The reference Mena (2010) is an unpublished thesis available at the University digital library. We have updated the reference to:

Mena, S. P.: Evolución de la dinámica de los escurrimientos en zonas de alta montaña: caso del Volcán Antisana (Unpublished dissertation), Thesis, Facultad de Ingeniería Civil y Ambiental, Escuela Politécnica Nacional, Quito - Ecuador. Retrieved from: http://bibdigital.epn.edu.ec/handle/15000/2503, 2010.

**19) A table is missing with the indication for all the water samples, locations, main characteristics, etc. . ..**

It has been included in the Annex section as advised in the referee's GC 2 (Comment No.4).

**20) Figure 1: The latitude and the longitude have to be added, the sources of data for each map have to be mentioned.**

Thanks for the comment. We have added the coordinates to the main map and added the sources of data for each one.

**21) Figure 2: What is the time step for the temperature data? Monthly? If it is the case please indicate "monthly temperature". If the temperatures are monthly temperatures, it would be better to plot the values with points without a line between them.**

Indeed, the values for temperature are average monthly maximum and average monthly minimum temperature. We have updated the Figure 2 accordingly.

[Figure]

Figure 2. Average monthly precipitation, maximum and minimum average monthly temperatures at Humboldt and Crespos weather stations from 2000 to 2011.

**22) Figure 3: How is made the separation between the sub-catchment, is it topographic? Which is the DEM (source, resolution) and which methodology has been used to separate each sub-catchment?**

We used a GIS-based subcatchment division approach to delineate the subcatchments for our study. We have updated the text in the manuscript on Pg 4 line 6: *"The DEM has a resolution of 20 x 20m and it was obtained from the contour line from the Ecuadorian Military Geographical Institute (IGM) scale 1:50000. The stream network was based on 'hydrological approach' as defined by Mark (1984) (Lo & Yeung, 2007) and later verified during ground-truthing recording GPS point measurements and field observations. Subcatchment delineation used the multiple flow direction model (Tarboton, 1997) and the eight-direction method (D8) introduced by O'Callaghan amd Mark (1984).*

In references:

Lo, C. P. and Yeung, A. K. W. (Eds.): Concept and techniques in geographic information systems, Second Edition. Prentice Hall, 2007.

Mark, D. M.: Automated detection of drainage networks from digital elevation models, Cartographica, 21, 168-178, 1984.

O' Callaghan, J. F. and Mark, D. M.: The extraction of drainage networks from digital elevation data, Comput. Vis. Graph. Image Process, 28, 328-344, 1984.

Tarboton, D. G.: A new method for the determination of flow directions and upslope areas in grid digital elevation models Water Resour. Res. , 33, 309–319, 1997.

**23) How to be sure that the springwaters are not superficial rivers (see GC 1)?**

One of the limitations of this study is that it was not possible to clearly identify the source of the springwater. As explained in GC 1 (Comment #2) the springwater could come from a shallow subsurface flow or from a deeper groundwater flow that comes to the surface via fissures and fractures of the rock. Springwater samples were tested for significant difference based on their geological background (please refer to the reply on Comment #3).

**24) Some sub-catchments are missing, for example 21, 31, 32. Please add these sub catchments to the figure.**

Those numbers were left out due to the lack of space. We have updated the Figure to include them.

**25) Figure 4: The font size is too small. How can you draw box plots for the categories "ice" and "precipitation" with only 3 and 4 values respectively?**

We have improved the font size of all figures. Regarding the boxplots for the categories "ice" and "precipitation" please refer to the explanation given above in Comment # 12.

**26) Concerning the δ18O and δ2H rates, how do you explain the differences between the two categories "Ice" and "Precipitation"?**

The difference has to do with the fractionation of isotopes. Precipitation has a "heavier isotopic signature" than ice because the heavy molecules are the ones that fall during a rainfall event while for ice and glacier is the lightest. The isotopic signature for the ice is influenced by several aspects, mainly the primary isotopic composition of water at its formation, and the isotopic fractionation during freezing, melting and sublimation processes.

**27) Figure 5: The font size is too small. It should be indicated that the numbers for the X-axis represent the number of the sub-catchment. How many samples are used to draw the different box-plots? The number n of samples has to be specify (may be in the new table).**

Yes, the number of samples are specified in the new table (Annex section). All figures have been improved.

**28) Figure 6: The font size is too small.**

Thanks for your comment. We have updated all the Figures to be clearer.

**29) Figure 7: see comment above (P12l.5-6).**

The deviation of the precipitation and ice samples from the GMWL and LMWL curves were explained above in the reply to comment # 14 and 15.

**30) Figure 8: Please be more precise and define the three following terms: "pre-event" "event" and "post-event"**

Thanks for the observation. We have updated the definition of those three terms in section 3.2.2 on Pg 5 line 15: *"Surface water samples at the outlet were collected with a resolution of 15-20 min at three different phases: 1) Pre-event, which are water samples taken before the peak of the rainfall event (n=3), 2) Event, samples taken during the rainfall event (n=13), and 3) Post-event, which are samples taken after the peak of the rainfall event (n=10)."*

**31) Figure 9: No mention is done to Qspringwater : why?**

Please refer to the reply on Comment #1.

Annex

**Table A-1.** Location of the water samples taken from four different runoff source (Ice, Prec = precipitation, SW = springwater and Surf = surface water) and main characteristics and chemical concentrations (EC = electrical conductivity [µS/cm], $SiO_2$ [mg/l], $Cl^-$ [mg/l], $SO_4^{2-}$ [mg/l], $Na^+$ [m/l], $Mg^{2+}$ [mg/l], $K^+$ [mg/l], $Ca^{2+}$ [mg/l], $\delta^2H$ [‰], $\delta^{18}O$ [‰]) taken during July 2014. The UTM coordinates (WGS84) of the area are Zone 17M North and East, Dist = distance to the outlet, Subcat = subcatchment, Cat = catchment and Geol = geological background (Ch = Chacana volcanic rocks, Hi = Hialina lava, LaRo = Lahar Rojo, Lapl = Lavas Pleistocene, Ti = Tillite late ice age). NA = samples are not available.

| ID | Source | N (m) | E (m) | Elev. | Dist. | EC | $SiO_2$ | $Cl^-$ | $SO_4^{2-}$ | $Na^+$ | $Mg^{2+}$ | $K^+$ | $Ca^{2+}$ | $\delta^2H$ | $\delta^{18}O$ | Subcat. | Cat. | Geol. |
|---|---|---|---|---|---|---|---|---|---|---|---|---|---|---|---|---|---|---|
| 10 | Ice | 9945370 | 816341 | 4736 | 7300 | NA | NA | 2.9 | 0.6 | 1.0 | 0.1 | 1.0 | 2.2 | -112.8 | -14.4 | 23 | 2 | - |
| 20 | Ice | 9945370 | 816341 | 4736 | 7300 | NA | NA | 1.3 | 0.4 | 1.0 | 0.0 | 0.1 | 1.0 | -111.6 | -14.2 | 23 | 2 | - |
| 30 | Ice | 9945370 | 816341 | 4736 | 7300 | NA | NA | 1.5 | 0.4 | 1.0 | 0.0 | 0.0 | 1.0 | -113.6 | -14.6 | 23 | 2 | - |
| P_01 | Prec | 9943248 | 810185 | 4060 | 1 | NA | 0.7 | 3.8 | 2.2 | 1.7 | 0.3 | 1.1 | 6.7 | -59.5 | -8.5 | 1 | 1 | - |
| P_02 | Prec | 9943248 | 810185 | 4060 | 1 | 7.9 | 0.9 | 1.3 | 0.7 | 1.0 | 0.0 | 0.1 | 1.4 | -45.4 | -5.7 | 1 | 1 | - |
| P_03 | Prec | 9943248 | 810185 | 4060 | 1 | 10.0 | 1.1 | 1.6 | 0.7 | 1.0 | 0.0 | 0.1 | 1.1 | -42.4 | -3.9 | 1 | 1 | - |
| P_04 | Prec | 9943248 | 810185 | 4060 | 1 | 35.4 | 0.5 | 2.6 | 1.1 | 1.2 | 0.1 | 0.9 | 2.0 | -58.4 | -5.7 | 1 | 1 | - |
| T_S1_01 | SW | 9943424 | 810258 | 4047 | 320 | 144.5 | 57.8 | 1.9 | 10.3 | 8.7 | 4.0 | 3.0 | 7.3 | -111.0 | -13.4 | 1 | 1 | Hi |
| T_S1_20 | SW | 9943451 | 812460 | 4166 | 2850 | 70.0 | NA | NA | NA | NA | NA | NA | NA | -89.1 | -11.9 | 7 | 5 | Ch |
| T_S1_26 | SW | 9943471 | 813147 | 4206 | 3600 | 116.2 | NA | NA | NA | NA | NA | NA | NA | -85.7 | -11.9 | 9 | 5 | Ti |
| T_S1_22 | SW | 9943489 | 812029 | 4135 | 2500 | NA | NA | NA | NA | NA | NA | NA | NA | -98.9 | -12.4 | 7 | 5 | LaPl |
| T_S1_25 | SW | 9943502 | 812928 | 4191 | 3370 | NA | NA | NA | NA | NA | NA | NA | NA | -90.4 | -12.7 | 9 | 5 | LaPl |
| T_S1_21 | SW | 9943521 | 812319 | 4146 | 2800 | 116.3 | NA | NA | NA | NA | NA | NA | NA | -99.6 | -13.6 | 7 | 5 | Ti |
| T_S1_16 | SW | 9943524 | 811802 | 4127 | 3670 | NA | NA | NA | NA | NA | NA | NA | NA | -89.5 | -11.9 | 5 | 5 | LaPl |
| T_S1_15 | SW | 9943526 | 811662 | 4122 | 3675 | 140.9 | NA | NA | NA | NA | NA | NA | NA | -105.2 | -14.4 | 5 | 5 | LaPl |
| T_S1_14 | SW | 9943543 | 811655 | 4121 | 2100 | 137.7 | NA | NA | NA | NA | NA | NA | NA | -106.7 | -14.6 | 5 | 5 | LaPl |
| T_S1_17 | SW | 9943547 | 812001 | 4134 | 2540 | 94.5 | 55.8 | NA | NA | 6.3 | 4.2 | 3.1 | 6.0 | -105.3 | -13.5 | 6 | 5 | Ch |
| T_S1_09 | SW | 9943576 | 811372 | 4109 | 2020 | 114.4 | NA | NA | NA | NA | NA | NA | NA | -88.4 | -12.5 | 5 | 5 | LaPl |
| T_S1_06 | SW | 9943599 | 811130 | 4114 | 1550 | 275.0 | 69.5 | 6.8 | 19.9 | 13.8 | 7.5 | 5.8 | 14.3 | -99.6 | -13.6 | 31 | 3 | LaRo |
| T_S1_02 | SW | 9943639 | 810532 | 4077 | 817 | 159.2 | NA | NA | NA | NA | NA | NA | NA | -109.4 | -14.6 | 1 | 1 | Hi |
| T_S1_03 | SW | 9943708 | 810640 | 4079 | 1005 | 140.0 | NA | NA | NA | NA | NA | NA | NA | -109.9 | -14.7 | 1 | 1 | Hi |
| T_S1_13 | SW | 9943739 | 810657 | 4083 | 1050 | 151.0 | 57.6 | 2.3 | 14.8 | 8.6 | 3.8 | 2.9 | 6.6 | -110.5 | -13.7 | 1 | 1 | Hi |

| ID | Source | N (m) | E (m) | Elev. | Dist. | EC | SiO$_2$ | Cl$^-$ | SO$_4^{2-}$ | Na$^+$ | Mg$^{2+}$ | K$^+$ | Ca$^{2+}$ | $\delta^2$H | $\delta^{18}$O | Subcat. | Cat. | Geol. |
|---|---|---|---|---|---|---|---|---|---|---|---|---|---|---|---|---|---|---|
| T_S1_04 | SW | 9943741 | 810709 | 4080 | 1113 | 330.0 | NA | NA | NA | NA | NA | NA | NA | -106.0 | -14.1 | 1 | 1 | LaRo |
| T_S1_29 | SW | 9943790 | 814275 | 4325 | 4850 | 86.6 | 49.9 | NA | NA | NA | NA | NA | NA | -98.6 | -13.1 | 9 | 5 | Ti |
| T_S1_12 | SW | 9943797 | 810758 | 4083 | 1100 | 339.0 | NA | NA | NA | NA | NA | NA | NA | -112.0 | -13.0 | 1 | 1 | LaRo |
| T_S1_27 | SW | 9943812 | 814257 | 4316 | 4780 | 99.1 | NA | 1.8 | 0.7 | 3.7 | 3.0 | 2.4 | 5.1 | -101.6 | -12.6 | 9 | 5 | Ti |
| T_S1_28 | SW | 9943825 | 814281 | 4319 | 4800 | 110.8 | 46.6 | NA | NA | NA | NA | NA | NA | -93.6 | -12.6 | 9 | 5 | Ti |
| T_S1_05 | SW | 9943878 | 811156 | 4100 | 1455 | 280.0 | 64.5 | 4.9 | 35.7 | 15.6 | 7.1 | 4.4 | 12.3 | -106.6 | -14.6 | 22 | 2 | LaRo |
| T_S1_31 | SW | 9943930 | 814545 | 4341 | 5015 | NA | NA | 1.5 | 1.4 | NA | NA | NA | NA | -94.9 | -11.6 | 9 | 5 | Ti |
| T_S1_32 | SW | 9943934 | 814574 | 4345 | 5060 | 59.0 | NA | NA | NA | NA | NA | NA | NA | -95.8 | -12.3 | 9 | 5 | Ch |
| T_S1_30 | SW | 9943935 | 814517 | 4334 | 4975 | 77.7 | NA | NA | NA | NA | NA | NA | NA | -101.6 | -13.1 | 9 | 5 | Ch |
| T_S1_33 | SW | 9943973 | 814685 | 4354 | 5170 | 58.8 | NA | NA | NA | NA | NA | NA | NA | -94.1 | -12.8 | 9 | 5 | Ch |
| T_S1_11 | SW | 9943975 | 811328 | 4103 | 1790 | 298.0 | 60.8 | 5.2 | 38.5 | 18.8 | 8.5 | 5.0 | 12.9 | -108.3 | -14.3 | 22 | 2 | LaRo |
| T_S1_34 | SW | 9943985 | 814740 | 4356 | 5200 | 67.0 | NA | NA | NA | NA | NA | NA | NA | -98.6 | -12.9 | 9 | 5 | Ch |
| T_S1_35 | SW | 9944017 | 814842 | 4369 | 5400 | 64.7 | NA | NA | NA | NA | NA | NA | NA | -97.7 | -13.2 | 9 | 5 | Ch |
| S1_49 | SW | 9944045 | 813496 | 4246 | 4150 | 75.5 | NA | 1.6 | 0.5 | 4.4 | 2.0 | 2.0 | 5.2 | -84.1 | -10.6 | 8 | 5 | Ti |
| T_S1_36 | SW | 9944054 | 814937 | 4382 | 5450 | 53.1 | NA | NA | NA | NA | NA | NA | NA | -100.4 | -12.8 | 9 | 5 | Ch |
| T_S3_14 | SW | 9944054 | 811454 | 4110 | 1950 | NA | 66.1 | NA | NA | NA | NA | NA | NA | -105.5 | -14.6 | 22 | 2 | Ch |
| T_S1_37 | SW | 9944066 | 814948 | 4386 | 5475 | 67.8 | NA | NA | NA | NA | NA | NA | NA | -105.2 | -13.7 | 9 | 5 | Ch |
| T_S3_13 | SW | 9944099 | 811526 | 4113 | 2100 | NA | NA | NA | NA | NA | NA | NA | NA | -107.2 | -14.8 | 22 | 2 | Ch |
| T_S1_38 | SW | 9944100 | 815040 | 4389 | 5530 | 69.5 | NA | NA | NA | NA | NA | NA | NA | -98.1 | -14.4 | 9 | 5 | Ch |
| T_S3_12 | SW | 9944108 | 811526 | 4113 | 2170 | 112.3 | NA | NA | NA | NA | NA | NA | NA | NA | NA | 22 | 2 | Ch |
| T_S3_11 | SW | 9944111 | 811570 | 4117 | 5980 | 103.5 | NA | NA | NA | NA | NA | NA | NA | -102.2 | -13.4 | 22 | 2 | Ch |
| T_S3_10 | SW | 9944237 | 811748 | 4129 | 2370 | 91.5 | 64.2 | 2.1 | 10.0 | 6.4 | 3.3 | 3.4 | 7.4 | -106.5 | -13.4 | 22 | 2 | Ch |
| T_S3_09 | SW | 9944239 | 811757 | 4129 | 4920 | 83.6 | NA | NA | NA | NA | NA | NA | NA | -107.3 | -13.2 | 22 | 2 | Ch |
| T_S3_02 | SW | 9944265 | 812357 | 4179 | 7100 | 63.2 | NA | NA | NA | NA | NA | NA | NA | -105.7 | -13.4 | 41 | 4 | Ch |
| T_S3_01 | SW | 9944332 | 812494 | 4190 | 6750 | 71.5 | NA | NA | NA | NA | NA | NA | NA | -104.7 | -13.6 | 41 | 4 | Ch |
| T_S3_08 | SW | 9944362 | 811920 | 4149 | 2590 | 97.6 | NA | NA | NA | NA | NA | NA | NA | -102.3 | -13.8 | 22 | 2 | Ch |
| T_S3_04 | SW | 9944462 | 812125 | 4171 | 3800 | 85.2 | 67.9 | 1.9 | 4.8 | 6.2 | 3.4 | 3.6 | 7.8 | -104.9 | -13.7 | 22 | 2 | Ch |
| T_S3_03 | SW | 9944477 | 812033 | 4159 | 2550 | 90.1 | 69.6 | 2.0 | 7.5 | 6.3 | 3.1 | 3.4 | 7.6 | -102.7 | -13.4 | 22 | 2 | Ch |
| T_S3_05 | SW | 9944521 | 812189 | 4176 | 3900 | 128.1 | 62.2 | 1.8 | 35.0 | 6.1 | 3.7 | 3.4 | 9.2 | -102.7 | -12.9 | 22 | 2 | LaPl |
| T_S3_06 | SW | 9944576 | 812602 | 4209 | 2550 | 67.1 | NA | NA | NA | NA | NA | NA | NA | -95.5 | -12.0 | 22 | 2 | Ch |
| T_S3_07 | SW | 9944602 | 813584 | 4274 | 2900 | 7.4 | NA | NA | NA | NA | NA | NA | NA | -35.9 | -4.2 | 22 | 2 | Ti |

| ID | Source | N (m) | E (m) | Elev. | Dist. | EC | $SiO_2$ | $Cl^-$ | $SO_4^{2-}$ | $Na^+$ | $Mg^{2+}$ | $K^+$ | $Ca^{2+}$ | $\delta^2H$ | $\delta^{18}O$ | Subcat. | Cat. | Geol. |
|---|---|---|---|---|---|---|---|---|---|---|---|---|---|---|---|---|---|---|
| S1_01 | Surf | 9943252 | 810178 | 4044 | 100 | 127.2 | 44.8 | 2.6 | 9.4 | 7.0 | 4.0 | 2.4 | 5.9 | -98.3 | -13.1 | 1 | 1 | - |
| S1_02 | Surf | 9943306 | 810262 | 4046 | 200 | 122.8 | NA | NA | NA | NA | NA | NA | NA | -101.3 | -12.1 | 1 | 1 | - |
| S1_03 | Surf | 9943396 | 810158 | 4047 | 300 | 124.4 | 45.5 | 2.3 | 7.7 | 7.4 | 4.1 | 2.6 | 6.4 | -101.7 | -12.0 | 1 | 1 | - |
| S1_04 | Surf | 9943443 | 810304 | 4053 | 400 | 112.5 | NA | NA | NA | NA | NA | NA | NA | -98.7 | -12.1 | 1 | 1 | - |
| S1_25 | Surf | 9943452 | 812467 | 4166 | 2950 | 36.3 | NA | NA | NA | NA | NA | NA | NA | -95.6 | -13.2 | 7 | 5 | - |
| S1_05 | Surf | 9943467 | 810390 | 4066 | 500 | 116.3 | 44.5 | 2.7 | 9.7 | 7.1 | 3.8 | 2.5 | 5.9 | -98.6 | -11.9 | 1 | 1 | - |
| T_S1_10 | Surf | 9943470 | 811664 | 4122 | 2240 | 72.0 | 38.0 | 2.2 | 2.1 | 2.5 | 2.2 | 1.5 | 3.6 | -98.6 | -12.6 | 7 | 5 | - |
| S1_27 | Surf | 9943471 | 812000 | 4133 | 2480 | 45.4 | NA | NA | NA | NA | NA | NA | NA | -97.5 | -12.5 | 7 | 5 | - |
| S1_24 | Surf | 9943477 | 812678 | 4179 | 3200 | 59.7 | 42.5 | 1.6 | 0.5 | 3.4 | 2.8 | 2.4 | 4.7 | -82.5 | -10.9 | 8 | 5 | - |
| S1_28 | Surf | 9943478 | 812864 | 4187 | 3300 | 57.2 | 37.6 | 1.7 | 0.9 | 2.4 | 1.5 | 1.6 | 3.0 | -96.6 | -13.4 | 9 | 5 | - |
| S1_29 | Surf | 9943481 | 813062 | 4200 | 3500 | 54.7 | NA | NA | NA | NA | NA | NA | NA | -95.3 | -13.4 | 9 | 5 | - |
| S1_06 | Surf | 9943505 | 810474 | 4076 | 600 | 110.9 | NA | NA | NA | NA | NA | NA | NA | -98.3 | -12.2 | 1 | 1 | - |
| S1_26 | Surf | 9943507 | 812253 | 4143 | 2750 | 43.7 | 40.3 | 1.6 | 0.8 | 2.0 | 1.5 | 1.3 | 3.1 | -98.2 | -12.5 | 7 | 5 | - |
| T_S1_19 | Surf | 9943514 | 812666 | 4179 | 2700 | 68.2 | NA | NA | NA | NA | NA | NA | NA | -82.5 | -10.9 | 8 | 5 | - |
| S1_18 | Surf | 9943529 | 811891 | 4129 | 2500 | 48.9 | NA | NA | NA | NA | NA | NA | NA | -104.5 | -13.9 | 6 | 5 | - |
| S1_07 | Surf | 9943548 | 810515 | 4078 | 700 | 124.5 | 46.2 | 2.7 | 9.9 | 7.8 | 4.2 | 2.8 | 6.3 | -100.0 | -12.5 | 1 | 1 | - |
| S1_16 | Surf | 9943548 | 811455 | 4112 | 2190 | 127.7 | NA | NA | NA | NA | NA | NA | NA | -102.4 | -14.2 | 5 | 5 | - |
| B_1 | Surf | 9943548 | 811455 | 4116 | 1850 | 128.7 | 38.8 | 2.4 | 6.3 | 5.5 | 2.8 | 2.0 | 5.1 | NA | NA | 31 | 3 | - |
| S1_19 | Surf | 9943557 | 812079 | 4139 | 2700 | 47.7 | 48.5 | 1.8 | 1.2 | 3.9 | 1.7 | 2.2 | 4.1 | -108.6 | -13.8 | 6 | 5 | - |
| S1_17 | Surf | 9943558 | 811675 | 4122 | 2300 | 52.9 | 45.6 | 1.9 | 1.4 | 4.1 | 2.1 | 2.1 | 3.9 | -102.3 | -13.5 | 6 | 5 | - |
| A_1 | Surf | 9943562 | 811460 | 4116 | 2000 | 254.8 | 57.8 | 6.4 | 15.7 | 10.5 | 6.0 | 4.9 | 18.5 | NA | NA | 32 | 3 | - |
| S1_23 | Surf | 9943566 | 812862 | 4189 | 3400 | 50.1 | NA | NA | NA | NA | NA | NA | NA | -82.8 | -11.5 | 8 | 5 | - |
| S1_30 | Surf | 9943569 | 813246 | 4214 | 3700 | 54.2 | 35.5 | 1.3 | 1.1 | 2.7 | 1.9 | 1.8 | 4.2 | -96.8 | -13.5 | 9 | 5 | - |
| S1_08 | Surf | 9943621 | 810526 | 4078 | 800 | 117.9 | NA | NA | NA | NA | NA | NA | NA | -99.1 | -12.5 | 1 | 1 | - |
| T_S1_23 | Surf | 9943624 | 812177 | 4149 | 2670 | 44.1 | NA | NA | NA | NA | NA | NA | NA | -107.8 | -13.7 | 6 | 5 | - |
| S1_15 | Surf | 9943630 | 811327 | 4107 | 1900 | 106.1 | 45.6 | 2.3 | 6.0 | 5.7 | 3.4 | 2.3 | 5.0 | -99.8 | -13.3 | 5 | 5 | - |
| T_S1_24 | Surf | 9943638 | 812201 | 4155 | 2680 | 43.5 | 47.5 | 1.9 | 1.3 | 2.8 | 1.3 | 1.6 | 2.8 | -108.3 | -13.6 | 6 | 5 | - |
| S1_22 | Surf | 9943661 | 813059 | 4204 | 3600 | 47.3 | 44.3 | 1.5 | 0.4 | 4.2 | 2.9 | 2.8 | 5.8 | -84.6 | -11.2 | 8 | 5 | - |
| S1_31 | Surf | 9943669 | 813421 | 4230 | 3900 | 52.3 | NA | NA | NA | NA | NA | NA | NA | -95.3 | -13.1 | 9 | 5 | - |
| 3_2C | Surf | 9943673 | 811237 | 4103 | 1620 | 142.0 | 42.3 | 1.8 | 3.9 | 6.7 | 3.4 | 2.3 | 5.4 | NA | NA | 31 | 3 | - |
| S3_17 | Surf | 9943675 | 811337 | 4108 | 2020 | 75.2 | NA | NA | NA | NA | NA | NA | NA | -103.1 | -12.4 | 41 | 4 | - |

| ID | Source | N (m) | E (m) | Elev. | Dist. | EC | SiO$_2$ | Cl$^-$ | SO$_4^{2-}$ | Na$^+$ | Mg$^{2+}$ | K$^+$ | Ca$^{2+}$ | $\delta^2$H | $\delta^{18}$O | Subcat. | Cat. | Geol. |
|---|---|---|---|---|---|---|---|---|---|---|---|---|---|---|---|---|---|---|
| S1_32 | Surf | 9943689 | 813619 | 4244 | 4100 | 52.8 | 35.8 | 1.5 | 1.0 | 2.9 | 1.8 | 1.9 | 4.1 | -99.2 | -13.0 | 9 | 5 | - |
| S1_09 | Surf | 9943690 | 810581 | 4078 | 900 | 120.0 | 45.5 | 2.8 | 10.1 | 7.0 | 3.5 | 2.6 | 5.6 | -100.4 | -12.4 | 1 | 1 | - |
| T_S1_08 | Surf | 9943703 | 811488 | 4116 | 1120 | 93.5 | 49.4 | 1.5 | 2.4 | 4.8 | 3.0 | 2.5 | 4.7 | -98.9 | -13.4 | 41 | 4 | - |
| S1_10 | Surf | 9943706 | 810637 | 4079 | 1000 | 130.0 | NA | NA | NA | NA | NA | NA | NA | -99.5 | -13.3 | 1 | 1 | - |
| S3_16 | Surf | 9943711 | 811540 | 4119 | 950 | 69.1 | 50.5 | 1.7 | 2.4 | 5.3 | 2.8 | 2.8 | 5.0 | -103.2 | -12.7 | 41 | 4 | - |
| C_1 | Surf | 9943717 | 811140 | 4114 | 1840 | 131.6 | 39.7 | 2.9 | 9.3 | 3.8 | 1.9 | 1.5 | 3.2 | NA | NA | 31 | 3 | - |
| S1_14 | Surf | 9943724 | 811178 | 4100 | 1500 | 144.9 | 42.7 | 3.2 | 6.5 | 6.7 | 3.4 | 2.3 | 8.1 | -101.9 | -13.5 | 31 | 3 | - |
| S3_15 | Surf | 9943733 | 811602 | 4123 | 1220 | 90.1 | 49.6 | 1.6 | 0.6 | 6.9 | 5.7 | 3.7 | 6.8 | -92.0 | -11.4 | 41 | 4 | - |
| S1_11 | Surf | 9943735 | 810706 | 4082 | 1100 | 110.0 | 45.3 | 2.9 | 9.5 | 6.9 | 3.7 | 2.6 | 5.9 | -101.0 | -12.2 | 1 | 1 | - |
| S1_12 | Surf | 9943746 | 810801 | 4088 | 1200 | 120.0 | NA | NA | NA | NA | NA | NA | NA | -99.6 | -13.1 | 1 | 1 | - |
| S1_33 | Surf | 9943756 | 813817 | 4259 | 4300 | 52.0 | NA | NA | NA | NA | NA | NA | NA | -98.7 | -12.8 | 9 | 5 | - |
| S1_13 | Surf | 9943767 | 810978 | 4092 | 1300 | 134.9 | 46.8 | 3.0 | 10.7 | 7.4 | 4.0 | 2.7 | 6.4 | -102.6 | -13.4 | 1 | 1 | - |
| 21_1_2C | Surf | 9943783 | 811014 | 4094 | 1345 | 247.2 | 59.7 | 1.5 | 19.1 | 11.4 | 5.9 | 3.3 | 10.5 | NA | NA | 21 | 2 | - |
| S1_34 | Surf | 9943790 | 914018 | 4281 | 4500 | 49.1 | 34.5 | 1.9 | 2.2 | 2.2 | 1.4 | 1.5 | 2.9 | -105.9 | -13.7 | 9 | 5 | - |
| S1_35 | Surf | 9943814 | 814221 | 4314 | 4700 | 50.6 | NA | NA | NA | NA | NA | NA | NA | -101.4 | -12.8 | 9 | 5 | - |
| S1_21 | Surf | 9943817 | 813187 | 4225 | 3800 | 46.3 | NA | NA | NA | NA | NA | NA | NA | -82.4 | -11.4 | 8 | 5 | - |
| S3_14 | Surf | 9943854 | 811772 | 4133 | 1545 | 91.5 | NA | NA | NA | NA | NA | NA | NA | -95.4 | -11.8 | 41 | 4 | - |
| S1_36 | Surf | 9943871 | 814412 | 4324 | 4900 | 53.3 | 35.9 | 1.4 | 0.7 | 2.5 | 1.5 | 1.6 | 3.4 | -103.6 | -13.1 | 9 | 5 | - |
| 22_1_2C | Surf | 9943878 | 811156 | 4050 | 1445 | 280.0 | 64.5 | 4.9 | NA | 15.6 | 7.1 | 4.4 | 12.3 | NA | NA | 22 | 2 | - |
| 21_2_2C | Surf | 9943894 | 811126 | 4100 | 1420 | 202.0 | 47.9 | 0.8 | 11.8 | 9.8 | 4.9 | 2.8 | 8.9 | NA | NA | 21 | 2 | - |
| S2_28 | Surf | 9943926 | 811107 | 4101 | 1500 | 11.8 | 10.6 | 1.3 | 0.8 | 1.0 | 0.3 | 0.4 | 1.2 | -89.9 | -12.5 | 23 | 2 | - |
| S1_20 | Surf | 9943934 | 813378 | 4236 | 4000 | 42.5 | 40.7 | 1.6 | 0.5 | 3.3 | 2.0 | 2.2 | 4.4 | -80.9 | -11.3 | 8 | 5 | - |
| S1_37 | Surf | 9943957 | 814637 | 4347 | 5100 | 49.9 | NA | NA | NA | NA | NA | NA | NA | -99.5 | -13.4 | 9 | 5 | - |
| S1_38 | Surf | 9944019 | 814821 | 4365 | 5300 | 49.0 | 32.7 | 1.6 | 1.3 | 1.5 | 0.9 | 1.0 | 3.7 | -96.7 | -13.7 | 9 | 5 | - |
| S3_13 | Surf | 9944046 | 811889 | 4146 | 1990 | 87.7 | 52.4 | 1.6 | 0.9 | 5.8 | 4.7 | 3.2 | 5.7 | -95.0 | -11.8 | 41 | 4 | - |
| S1_39 | Surf | 9944090 | 815015 | 4386 | 5500 | 46.1 | NA | NA | NA | NA | NA | NA | NA | -96.0 | -13.5 | 9 | 5 | - |
| S2_29 | Surf | 9944102 | 811177 | 4127 | 1700 | 9.4 | 10.5 | 1.6 | 1.0 | 1.3 | 0.3 | 0.5 | 1.5 | -90.0 | -12.4 | 23 | 2 | - |
| S3_12 | Surf | 9944139 | 812070 | 4161 | 2190 | 71.3 | NA | NA | NA | NA | NA | NA | NA | -104.9 | -13.2 | 41 | 4 | - |
| S1_40 | Surf | 9944153 | 815208 | 4410 | 5700 | 41.1 | 27.2 | 1.4 | 1.6 | 1.5 | 1.0 | 1.0 | 2.6 | -96.5 | -13.7 | 9 | 5 | - |
| S3_11 | Surf | 9944207 | 812233 | 4172 | 2370 | 73.4 | 49.5 | 1.9 | 0.6 | 3.9 | 2.7 | 3.1 | 5.0 | -98.4 | -12.3 | 41 | 4 | - |
| S2_27 | Surf | 9944207 | 811300 | 4138 | 1900 | 10.2 | NA | NA | NA | NA | NA | NA | NA | -91.5 | -12.3 | 23 | 2 | - |

| ID | Source | N (m) | E (m) | Elev. | Dist. | EC | $SiO_2$ | $Cl^-$ | $SO_4^{2-}$ | $Na^+$ | $Mg^{2+}$ | $K^+$ | $Ca^{2+}$ | $\delta^2H$ | $\delta^{18}O$ | Subcat. | Cat. | Geol. |
|---|---|---|---|---|---|---|---|---|---|---|---|---|---|---|---|---|---|---|
| S1_41 | Surf | 9944209 | 815393 | 4431 | 5900 | 33.9 | NA | NA | NA | NA | NA | NA | NA | -99.8 | -13.1 | 9 | 5 | - |
| S1_46 | Surf | 9944216 | 816351 | 4596 | 7000 | 23.5 | 11.4 | 2.1 | 3.2 | 1.2 | 0.5 | 0.7 | 2.3 | -100.8 | -13.7 | 9 | 5 | - |
| S1_45 | Surf | 9944223 | 816189 | 4567 | 6800 | 24.8 | NA | NA | NA | NA | NA | NA | NA | -99.8 | -12.5 | 9 | 5 | - |
| S1_44 | Surf | 9944233 | 816003 | 4538 | 6600 | 26.5 | 12.5 | 2.1 | 2.7 | 1.3 | 0.6 | 0.8 | 2.4 | -96.8 | -13.0 | 9 | 5 | - |
| S1_47 | Surf | 9944273 | 816511 | 4634 | 7200 | 22.9 | NA | NA | NA | NA | NA | NA | NA | -98.9 | -13.4 | 9 | 5 | - |
| S1_42 | Surf | 9944323 | 815567 | 4462 | 6100 | 32.5 | 19.8 | 1.3 | 2.0 | 1.4 | 0.8 | 1.0 | 2.7 | -97.7 | -13.9 | 9 | 5 | - |
| S3_08 | Surf | 9944327 | 812892 | 4221 | 7000 | 41.0 | NA | NA | NA | NA | NA | NA | NA | -95.5 | -11.6 | 41 | 4 | - |
| T_S1_39 | Surf | 9944328 | 815945 | 4522 | 6550 | 39.2 | 25.7 | NA | NA | 2.2 | 0.9 | 1.3 | 2.8 | -108.0 | -13.7 | 9 | 5 | - |
| S3_09 | Surf | 9944331 | 812703 | 4207 | 6600 | 39.2 | 48.4 | 1.7 | 1.1 | 2.7 | 1.4 | 1.7 | 2.2 | -100.3 | -12.3 | 41 | 4 | - |
| S3_10 | Surf | 9944331 | 812495 | 4190 | 7300 | 70.5 | NA | NA | NA | NA | NA | NA | NA | -102.5 | -13.2 | 41 | 4 | - |
| S1_43 | Surf | 9944356 | 815746 | 4492 | 6300 | 31.8 | NA | NA | NA | NA | NA | NA | NA | -97.3 | -13.7 | 9 | 5 | - |
| S1_48 | Surf | 9944382 | 816692 | 4673 | 7400 | 21.3 | 7.5 | 1.4 | 3.6 | 1.0 | 0.6 | 0.6 | 3.1 | -100.9 | -12.8 | 9 | 5 | - |
| S2_26 | Surf | 9944402 | 811469 | 4148 | 2100 | 10.3 | 6.2 | 1.3 | 1.0 | 1.8 | 0.3 | 0.6 | 1.9 | -90.7 | -12.4 | 23 | 2 | - |
| S3_07 | Surf | 9944403 | 813078 | 4236 | 3700 | 55.4 | 46.2 | 1.9 | 1.3 | 2.1 | 1.8 | 1.2 | 2.2 | -100.6 | -12.2 | 41 | 4 | - |
| S3_06 | Surf | 9944505 | 813280 | 4257 | 3900 | 64.7 | NA | NA | NA | NA | NA | NA | NA | -101.4 | -12.9 | 43 | 4 | - |
| S3_18 | Surf | 9944534 | 813410 | 4268 | 4030 | 55.1 | 44.6 | 1.6 | 0.6 | 3.3 | 3.2 | 2.0 | 4.3 | -92.4 | -12.1 | 42 | 4 | - |
| S3_19 | Surf | 9944594 | 813565 | 4286 | 4230 | 32.9 | NA | NA | NA | NA | NA | NA | NA | -99.7 | -13.1 | 42 | 4 | - |
| S2_25 | Surf | 9944600 | 811518 | 4157 | 2300 | 10.8 | NA | NA | NA | NA | NA | NA | NA | -88.7 | -12.3 | 23 | 2 | - |
| S3_20 | Surf | 9944666 | 813704 | 4297 | 4430 | 51.2 | 62.8 | 2.0 | 1.2 | 4.1 | 1.8 | 2.8 | 4.3 | -97.0 | -15.7 | 42 | 4 | - |
| S3_01 | Surf | 9944677 | 813367 | 4281 | 4100 | 68.7 | 54.1 | 1.6 | 1.7 | 2.3 | 2.7 | 1.4 | 2.8 | -105.6 | -12.9 | 43 | 4 | - |
| S3_21 | Surf | 9944713 | 813880 | 4314 | 4630 | 30.7 | NA | NA | NA | NA | NA | NA | NA | -103.7 | -12.7 | 42 | 4 | - |
| S3_02 | Surf | 9944766 | 813536 | 4303 | 4300 | 73.8 | NA | NA | NA | NA | NA | NA | NA | -104.8 | -12.9 | 43 | 4 | - |
| S2_24 | Surf | 9944809 | 811679 | 4170 | 2500 | 9.8 | 11.5 | 1.5 | 0.9 | 1.4 | 0.2 | 0.4 | 1.1 | -87.9 | -12.7 | 23 | 2 | - |
| S3_22 | Surf | 9944829 | 814097 | 4343 | 4830 | 45.8 | 53.2 | 1.8 | 1.6 | 4.0 | 2.0 | 2.6 | 4.9 | -110.8 | -14.0 | 42 | 4 | - |
| S3_03 | Surf | 9944918 | 813657 | 4317 | 4500 | 69.5 | 58.3 | 1.7 | 1.4 | 3.7 | 3.1 | 2.6 | 4.0 | -105.7 | -13.3 | 43 | 4 | - |
| S2_23 | Surf | 9944947 | 811791 | 4179 | 2700 | 9.5 | NA | NA | NA | NA | NA | NA | NA | -88.5 | -12.9 | 23 | 2 | - |
| S2_01 | Surf | 9945081 | 811901 | 4188 | 2900 | 7.6 | 12.8 | 1.2 | 0.8 | 1.0 | 0.3 | 0.3 | 1.5 | -93.0 | -13.0 | 23 | 2 | - |
| S3_04 | Surf | 9945090 | 813820 | 4341 | 4700 | 68.2 | NA | NA | NA | NA | NA | NA | NA | -107.8 | -13.8 | 43 | 4 | - |
| S2_02 | Surf | 9945155 | 812102 | 4208 | 3100 | 6.6 | NA | NA | NA | NA | NA | NA | NA | -93.5 | -13.0 | 23 | 2 | - |
| S3_05 | Surf | 9945164 | 813983 | 4370 | 4900 | 87.4 | 54.7 | 1.7 | 0.9 | 3.1 | 5.1 | 3.1 | 5.2 | -106.5 | -13.5 | 43 | 4 | - |
| S2_03 | Surf | 9945262 | 812266 | 4232 | 3300 | 6.7 | 11.2 | 1.3 | 0.9 | 1.2 | 0.2 | 0.4 | 1.4 | -95.2 | -12.9 | 23 | 2 | - |

| ID | Source | N (m) | E (m) | Elev. | Dist. | EC | $SiO_2$ | $Cl^-$ | $SO_4^{2-}$ | $Na^+$ | $Mg^{2+}$ | $K^+$ | $Ca^{2+}$ | $\delta^2H$ | $\delta^{18}O$ | Subcat. | Cat. | Geol. |
|---|---|---|---|---|---|---|---|---|---|---|---|---|---|---|---|---|---|---|
| Lake | Surf | 9945276 | 815577 | 4664 | 7050 | 6.6 | 3.9 | 1.2 | 0.7 | 1.6 | 0.2 | 0.5 | 1.0 | -87.6 | -12.6 | 23 | 2 | - |
| S2_22 | Surf | 9945276 | 815578 | 4664 | 7050 | 6.6 | 3.9 | 1.4 | 0.8 | 1.0 | 0.2 | 0.3 | 1.0 | NA | NA | 23 | 2 | - |
| S2_21 | Surf | 9945331 | 815528 | 4650 | 6900 | 6.3 | 3.7 | 1.7 | 0.8 | NA | NA | NA | NA | -94.9 | -13.4 | 23 | 2 | - |
| S2_04 | Surf | 9945351 | 812415 | 4237 | 3500 | 6.7 | NA | NA | NA | NA | NA | NA | NA | -94.1 | -12.9 | 23 | 2 | - |
| S2_05 | Surf | 9945411 | 812613 | 4260 | 3700 | 6.5 | 13.3 | 1.2 | 0.8 | 1.2 | 0.2 | 0.3 | 1.5 | -94.2 | -12.8 | 23 | 2 | - |
| S2_20 | Surf | 9945449 | 815378 | 4597 | 6700 | 6.4 | NA | NA | NA | NA | NA | NA | NA | -95.5 | -13.4 | 23 | 2 | - |
| S2_15 | Surf | 9945503 | 814384 | 4431 | 5700 | 6.9 | 5.6 | 1.2 | 0.8 | 1.3 | 0.3 | 0.4 | 1.5 | -97.4 | -13.1 | 23 | 2 | - |
| S2_19 | Surf | 9945504 | 815186 | 4552 | 6500 | 6.4 | 9.7 | 1.6 | 0.8 | 1.0 | 0.2 | 0.2 | 1.0 | -93.0 | -13.4 | 23 | 2 | - |
| S2_17 | Surf | 9945509 | 814813 | 4489 | 6100 | 6.6 | 6.3 | 1.2 | 0.8 | 1.0 | 0.2 | 0.3 | 1.5 | -97.2 | -12.9 | 23 | 2 | - |
| S2_14 | Surf | 9945510 | 814192 | 4413 | 5500 | 6.9 | NA | NA | NA | NA | NA | NA | NA | -96.9 | -13.2 | 23 | 2 | - |
| S2_18 | Surf | 9945524 | 814995 | 4521 | 6300 | 7.5 | NA | NA | NA | NA | NA | NA | NA | -95.8 | -12.8 | 23 | 2 | - |
| S2_06 | Surf | 9945534 | 812806 | 4275 | 3900 | 6.5 | NA | NA | NA | NA | NA | NA | NA | -95.0 | -12.7 | 23 | 2 | - |
| S2_16 | Surf | 9945534 | 814618 | 4460 | 5900 | 6.7 | NA | NA | NA | NA | NA | NA | NA | -96.6 | -12.9 | 23 | 2 | - |
| S2_07 | Surf | 9945599 | 813004 | 4294 | 4100 | 6.5 | 10.1 | 1.5 | 0.8 | 1.5 | 0.2 | 0.5 | 1.6 | -96.0 | -12.6 | 23 | 2 | - |
| S2_13 | Surf | 9945612 | 813989 | 4397 | 5300 | 6.9 | 9.5 | 1.2 | 0.8 | 1.2 | 0.2 | 0.4 | 1.4 | -97.2 | -13.1 | 23 | 2 | - |
| S2_12 | Surf | 9945716 | 813831 | 4370 | 5100 | 7.0 | NA | NA | NA | NA | NA | NA | NA | -94.0 | -12.4 | 23 | 2 | - |
| S2_08 | Surf | 9945758 | 813117 | 4303 | 4300 | 6.4 | NA | NA | NA | NA | NA | NA | NA | -94.0 | -12.6 | 23 | 2 | - |
| S2_11 | Surf | 9945769 | 813624 | 4353 | 4900 | 6.9 | 13.1 | 1.2 | 0.8 | NA | NA | NA | NA | -94.6 | -12.5 | 23 | 2 | - |
| S2_10 | Surf | 9945858 | 813448 | 4338 | 4700 | 6.7 | NA | NA | NA | NA | NA | NA | NA | -93.3 | -12.4 | 23 | 2 | - |
| S2_09 | Surf | 9945899 | 813229 | 4320 | 4500 | 6.5 | 11.5 | 1.3 | 0.8 | 1.1 | 0.1 | 0.3 | 1.3 | -93.8 | -12.5 | 23 | 2 | - |

**Table A2.** Statistical summary of the chemical components and stable istopes of water samples within the Los Crespos-Humboldt basin.

| | EC | $SiO_2$ | $Cl^-$ | $SO_4^{2-}$ | $Na^+$ | $Mg^{2+}$ | $K^+$ | $Ca^{2+}$ | $\delta^2H$ | $\delta^{18}O$ |
|---|---|---|---|---|---|---|---|---|---|---|
| *Ice (n=3)* | | | | | | | | | | |
| mean | - | - | 1.9 | 0.5 | 1.0 | 0.0 | 0.3 | 1.3 | -112.7 | -14.4 |
| std | - | - | 0.9 | 0.1 | 0.0 | 0.1 | 0.5 | 0.6 | 1.0 | 0.2 |
| min | - | - | 1.3 | 0.4 | 1.0 | 0.0 | 0.0 | 1.0 | -113.6 | -14.6 |
| max | - | - | 2.9 | 0.6 | 1.0 | 0.1 | 1.0 | 2.2 | -111.6 | -14.2 |
| *Precipitation (n=4)* | | | | | | | | | | |
| mean | 17.8 | 0.8 | 2.3 | 1.2 | 1.2 | 0.1 | 0.6 | 2.8 | -51.4 | -5.9 |
| std | 15.3 | 0.3 | 1.1 | 0.7 | 0.3 | 0.1 | 0.5 | 2.6 | 8.8 | 1.9 |
| min | 7.9 | 0.5 | 1.3 | 0.7 | 1.0 | 0.0 | 0.1 | 1.1 | -59.5 | -8.5 |
| max | 35.4 | 1.1 | 3.8 | 2.2 | 1.7 | 0.3 | 1.1 | 6.7 | -42.4 | -3.9 |
| *Springwater (n=46)* | | | | | | | | | | |
| mean | 119.9 | 61.0 | 2.8 | 14.9 | 8.7 | 4.5 | 3.5 | 8.5 | -99.5 | -13.1 |
| std | 77.7 | 7.2 | 1.7 | 14.2 | 4.8 | 2.1 | 1.1 | 3.1 | 12.0 | 1.6 |
| min | 7.4 | 46.6 | 1.5 | 0.5 | 3.7 | 2.0 | 2.0 | 5.1 | -112.0 | -14.8 |
| max | 339.0 | 69.6 | 6.8 | 38.5 | 18.8 | 8.5 | 5.8 | 14.3 | -35.9 | -4.2 |
| *Surface water (n=113)* | | | | | | | | | | |
| mean | 59.6 | 34.1 | 1.9 | 3.2 | 3.8 | 2.2 | 1.8 | 4.2 | -97.5 | -12.8 |
| std | 53.9 | 18.2 | 0.9 | 4.1 | 3.0 | 1.8 | 1.1 | 2.9 | 6.0 | 0.7 |
| min | 6.3 | 3.7 | 0.8 | 0.4 | 1.0 | 0.1 | 0.2 | 1.0 | -110.8 | -15.7 |
| max | 280.0 | 64.5 | 6.4 | 19.1 | 15.6 | 7.1 | 4.9 | 18.5 | -80.9 | -10.9 |

---

## Author Comment (AC2) · 14 Apr 2017

**Anonymous Referee #2:**

**The submission deals with runoff generation and water sources in a high-mountain catchments dominated by a glacier and characteristic Andean grasslands (paramo). The work reports sampling of hydro chemicals and stable water isotopes along the stream and for one event at the catchment outlet. This data is supported by precipitation, ice, and spring sampling. The approach is not novel, yet the landscape and the crucial role of grasslands makes this study of interest for the scientific community, especially as high-mountain and glaciated catchments are under increased and constant stress. Yet, I wished that the senior authors would have invested more time in their student's submission rather than passing the work on to the reviewers. I feel that the paper needs quite substantial revisions before it can be accepted. I think that this work**

We acknowledge and thank the contribution of the referee through his/her comments. Hereby we present a point-by-point reply to the referee´s questions and comments.

**MAJOR CONCERNS**

1) **i) What I am most critical about is the lack of a clear story line; especially the result section offers a bunch of data that were presented rather unstructured and it was hard to tease out the important bits of it. Why is what kind of data presented? I often felt lost, maybe also due to the lack of a proper description of the sampling.**

The authors' intention in the manuscript was to make a synthesis of the work carried out in the studied catchment by: 1) giving an overview of the hydrochemical catchment characterization, and identifying the main runoff sources and 2) determining the contribution of glacier and páramo runoff to total runoff during dry and wet conditions using stable isotopes.

We will improve the text in the data collection section 3.1, and we will add a table in the Annex section with the main characteristics, location and chemical composition of all the samples collected as also suggested by Reviewer 1 (please find the table in the Annex section of this response. The overall structure of the manuscript will be revised to make it more clear to the reader.

2) **ii) I felt confused throughout the read. There was plenty of sampling, along the stream, for events etc. What is clearly needed is a table and a more detailed description of the sampling.**

We acknowledge that a more detail description of all the samples will help to better understand the sampling done in the catchment study. This table has been added in the Annex section at the end of the manuscript.

3) **iii) I felt that the use of the sampling along the stream was never really detailed used and the core of the works ends up as one hydrograph separation (which lacks the proper methodological description). What precipitation was used as end-member etc.?**

We agree with the reviewer's comment that the main focus of the research is the estimation of the contribution of the glacier and paramo runoff during dry and wet conditions. However, the sampling along the stream assisted in understanding the spatial hydrochemical distribution and the various runoff sources leading to the identification of glacier and páramo runoff. Sampling along the stream also assisted in determining the most suitable tracers to identify these runoff contributions. In addition, sampling along the stream contributed with additional information of what is happening in specific areas. For instance, the outliers of the springwater samples provided evidenced of meltwater infiltration of the glacier, which requires more attention in a future study.

The precipitation samples collected for the same rainfall event were used for the end-member analysis and hydrograph separation.

4) **iv) I am somewhat critical about the lack of precipitation samples, especially since they show quite some spread and plot below the LMWL. How does the station where the LMWL was recorded compare to the local conditions (such as elevation etc.). This could lead to a clear offset of the true local LMWL. There is no clear description where ice was sampled. Three samples is somewhat small, but at least the spread is small too.**

Thank you for the comment. This point was also raised by Reviewer 1. The amount of precipitation samples depended on the amount of rainfall collected during the event.

During the analysis, these precipitation samples were noted for deviating from the LMWL and GMWL slopes. We hypothesize that the distance of the precipitation samples to the LMWL and GMWL might indicate secondary evaporation processes that occur when raindrops fall in a warm atmosphere. This is suggested by the values of deuterium excess less than 10‰ in all precipitation samples. The event was not a heavy rain, and therefore the raindrops are slightly more enriched.

A statement has been included in the manuscript with the possible explanation. However, an in-depth and further analysis is not within the scope of this study. Pg 16 Line 16: *"The isotopic composition of rainfall and their relative distance to the GWML propose a possible evaporation effect. The first raindrops are usually more isotopically enriched (Gat & Carmi, 1970). For the specific case of precipitation, further research on rainfall events at this location should be done to check for possible re-evaporation processes and contributions of different water vapor sources that might occur taking into account inter and intra event variability in the hydrological process."*

In References:

  Gat, J. R. and Carmi, I.: Evolution of the isotopic composition of atmospheric waters in the Mediterranean Sea area, Journal of Geophys. Res., 96, 13179-13188, 1970.

Izobamba station (0.37°S, 78.55°W) where the LMWL was recorded lies nearby the location where the samples were taken (0.5°S, 78.18°W). However, the conditions are slightly different. In terms of elevation the difference between Izobamba (3059 m a.s.l.) and

Humboldt (4010 m a.s.l.) is around 950m. The precipitation registered at Izobamba station is around 1400 mm/yr compared to 800 mm/yr at Humboldt station. The Andean Region is dominated by the altitude effect and could explain locally the significant deviations from the LWML.

We will include a statement to indicate where the ice samples were taken. We are aware that the amount of data is a limitation to make a complete analysis and determine the chemical signature of ice. We agree that the evidence is not conclusive of the chemical signature of ice based on the small dataset. In this regard, the results were only used to have an initial overview of the chemical components and stable isotopes present when comparing four different runoff sources including ice, precipitation, surface water and springwater.

5) **v) The landscape units are poorly described in the manuscript, e.g. Figure 1 does not even present the paramos.**

Pg 2 lines 39-41 describe briefly the dominant growth forms of the paramo vegetation. We will improve the legend of the land cover map in Figure 1c to clarify that the 3 growth forms are the páramo vegetation.

6) **vi) The figures quality in general. Figures 1-3 are acceptable quality, figures 4-9 are not.**

All figures will be improved in the revised version of this manuscript.

**MINOR CONCERNS:**

7) **The English could use a revision. The sentences are often nested and overly long. The title is not precise and too long. Please check the proper use of altitude, I think the differentiation between altitude and elevation needs some attention, cf. McVicar and Körner (2013), Oecologia. "On the use of elevation, altitude, and height in the ecological and climatological literature"**

We are grateful for the suggestion, and we will check the English language throughout the manuscript in the next version. The title has been changed to: "*Quantification of runoff generation from a combined glacier and paramo catchment during dry and wet conditions using environmental isotopes within the Antisana Ecological Reserve in the Ecuadorian highlands*" as also suggested by the first reviewer.

The paper from McVicar and Körmer (2013) gave a clear explanation of the different terms to explain vertical distance. We will consider changing altitude and altitudinal gradient for elevation and elevation gradient, respectively.

8) **The abstract needs some rework. Paramo is never explained. L14-16 lacks precision. The results need to be more detailed.**

The abstract has been updated incorporating the suggestions from reviewer 1 and reviewer 2.
Abstract:

"*In the Andean region, tropical grasslands more known as páramos are vitally important to serve the water needs of communities in the surrounding areas. Previous studies in combined glacier and páramo catchments have shown that the melting of glaciers contributes to runoff*

*generation and that the páramo ecosystem acts as a natural sponge, which plays an important role in regulating the runoff during the dry-season. However, not all runoff processes are well-understood in the Andean Region due to the high spatial variability of precipitation. Particularly in the northern Ecuador, young volcanic ash soil properties, soil moisture dynamics and other local factors such as vegetation interception and high radiation might influence the hydrological behavior. In addition, there is a lack of evidence of the origin and quantification of the contribution of runoff components in the páramo ecosystem. This study focuses on data collection and experimental investigations in a small catchment (15.2 km$^2$) that ranges between 4000 to 5700 m a.s.l. within the Antisana Ecological Reserve in the Ecuadorian Andean Region. It consists of 2.3 km$^2$ glaciers, 10.3 km$^2$ páramo grasslands and 2.6 km$^2$ moraines. The approach consists of the identification of suitable environmental tracers and hydrochemical features to identify the various runoff sources in order to determine their respective contribution during dry and wet conditions. Dry conditions defined as periods in which precipitation was absent for at least three consecutive days and wet conditions during rainfall events. The results show the great importance of the páramo on the contribution to total runoff during baseflow estimated around 70% and the capacity to dissipate the stream energy and buffer the peak flow during rainfall conditions. Electrical conductivity (EC) together with the stable isotopes ($\delta^{18}O$, and $\delta^2H$) were identified as conservative tracers that characterize the end-member concentrations."*

9) **P1L33-34: ". . .river's. . ..resurgence" this is unclear.**

   This sentence has been revised: *"... that feed directly the river's drainage system or may contribute further downstream due to glacial meltwater infiltrations and water resurgence through springs (Cauvy-Fraunié et al., 2013; Favier et al., 2008; Villacis, 2008)"*

10) **P1L39. The concern. . .. Check the grammar.**

    Thank you for the observation. We will correct the grammar in the new version.

11) **P2L10ff: Please make the research gap clearer, really try to make an effort why exactly this study is needed.**

    Pg2 lines 12-14 were revised to address this comment: *"However, appropriate tracers for a suitable spatial hydrochemical characterization have not been yet identified. In addition, a quantification of the different contributions from glacier and páramo components in these catchments of complex geology and topography remains a challenge."*

12) **P2L15ff: Restructure and specify the research question (or use hypotheses).**

    This paragraph has been revised: *"This study aims to i) identify effective environmental tracers (stable isotopes and major ions) to quantify the contribution of the main runoff components, and ii) provide a fair understanding of the hydrological interactions of a glacierized-páramo system during dry and wet conditions."*

**13) P2L33: "Location". For me this seems not right, as there is much more described here, such as catchment properties.**

Thank you for the observation. We have included subtitles for the paragraphs. Before line 38: Land cover and before line 44: Soil properties.

**14) P2L34ff. It was not always clear to me if the individual information was for the catchment, or the region. Furthermore, I found it difficult to follow since the authors jumped around in there references to distinct elevation ranges. What is low, what is high, etc. Please consider restructuring the section 2.**

Our intention was to differentiate the catchment properties between different elevations as described in previous studies that were linked to this one. However, section 2 will be updated. The following changes have been made:

*"Location*

*The catchment study Los Crespos-Humboldt (15.2 km2) lies within the Antisana Ecological Reserve (628.1 km2) in the Andean region of Ecuador (Figure 1a). It is located at the south-western slope of the Antisana volcano (0°30'S, 78°11'W) and its elevation ranges from 4000 to 5700 m a.s.l. This catchment is one of several water sources for La Mica reservoir that supplies water for the southern part of Quito, the capital city of Ecuador, located 50 km north of this catchment.*

*Land cover*

*It consists of 15% glaciers, 68% páramo grasslands and 17% moraines (Figure 1b). The latter one is an ecosystem in transition between the glacier and the páramo. The páramo vegetation is dominated by tussock grasses (TU) (Calamagrostis intermedia), acaulescent rosettes (AR) (Werneria nubigena, Hypochaeris sessiliflora) and cushions (CU) (Azorella Pedunculata) (Minaya et al., 2015) (Figure 1c). The páramo vegetation has adapted to specific climatic conditions of low atmospheric pressure, high radiation and wind drying effects (Luteyn, 1999). The glacier is an icecap that has retreated around 200 meters in the last 20 years (Cáceres et al., 2005; Hall et al., 2012).*

*Soil properties*

*Soils are mainly andosols, based on the FAO classification (Gardi et al., 2014), derived from volcanic material characterized by their high soil moisture (Buytaert et al., 2005a) and water retention capacity (Janeau et al., 2015; Roa-García et al., 2011). The soil texture varies from silty loam at elevations between 4000 to 4400 m a.s.l. to sandy clay loam soil at higher elevations where vegetation is sparse (Minaya et al., 2015). The soil texture influences the ecological and hydrological processes. Sandy soils drain well and reduce the capability of holding moisture; silty soils offer a high water-holding capacity. The slopes are moderate (up to 15°) at lower elevations and increases up to 30° close to the moraines at higher elevations."*

**15) P2L45-47: Not sure if some information that is clearly not linked to the catchment is needed.**

We agree that some information might not be needed. Careful revision will be made of this section to make it more concise.

**16) P4L10-11. Make it easier. Subcatchment#2 flows through. . .. and subcatchment#9. . .**

Thank you for the suggestion. The sentence has been updated to: *"Subcatchment # 2 flows through large boulders and rocks of different size; whilst subcatchment 9 flows through páramo vegetation."*

**17) P5L4: Please introduce a table where details are described.**

A table with the main characteristics and location of the samples has been added in the Annex section in the revised manuscript. It is also added at the end of this response.

**18) P5L5: Every 200m, you leave it for the reader to guess. You should clearly state that this was sampling along the flowing stream channel.**

Thank you for the comment. The sentence has been updated to: *"Every 200m along the flowing stream channel, electrical conductivity (EC) and...."*

**19) P5L10: reference for the method P5L14: You mean catchment outlet? How many events?**

The reference for the method has been included:

HACH: Silica, Silicomolybdate HR Method 8185, Powder Pillows. DOC316.53.01133. Edition 9. Hach Company/Hach Lange GmbH, 2014.

Sentence on Pg5 line 14 has been updated to: *"During rainfall events, the sampling of the surface water and precipitation was undertaken only at the outlet of the catchment at Humboldt Station for rainfall-runoff analyses. Samples were collected with a resolution of 15-20 min during the rainfall event."*

**20) P5L16: When were the precip samples taken?**

This information has been added in Section 3.1 to clarify that the period is different for dry and wet conditions, pg4 line 8: *"Isotopic and hydrochemical samples were collected in a sampling campaign carried out in July 2014. For dry conditions July 4-7 and for wet conditions July 14-15."*

**21) P5L29ff: The text would profit when transformed to active style. The PCA was not mentioned in the methods.**

We have deleted the sentence of PCA on Pg 10 lines 3-4 to address this comment.

**22) P6L1: The header is unclear, I am also not sure what exactly was done with this mixing analysis. I stay confused.**

We have changed the header to Flow routing to reflect the description in section 3.4.2.

Regarding the mixing analysis, the intention was to identify interrelationships between major ions using the chemical components and find their suitability as tracers for the hydrograph separation.

**23) P6L29: Maybe "end member" would be a better term?**

We agree that subtitle 4.1.1 Runoff sources does not reflect the content of this subsection. Therefore we will delete this subtitle, and this section will fall under the Hydrochemical catchment characterization description.

**24) P6L30ff.: See major concerns. Please avoid methods in the results, describe everything in section 3.**

We agree that it is repetitive. In this regard, the first sentence has been deleted.

**25) P7: Figure 4: Box plots with 3 and 4 data points are sketchy. The quality of the figure needs to be improved, line width, font size, a, b etc is not mentioned in the captions**

This comment was also raised by reviewer 1. We are aware of the limited number of ice and precipitation samples. We have updated Figure 4 and replaced the box-plots for other type of plot showing only the distribution of samples and their mean.

[Figure]

Figure 4. Chemical components and stable isotopes of water samples within the Los Crespos-Humboldt basin of different runoff sources (Ice, Precipitation, Surface and Spring water). The blue dots represent the samples and the red star the mean values. NA= samples are not available.

**26) P8L2: Avoid introductory sentences in the results paragraph. Straight to the point. What is important?**

Thanks for the comment. It will be taken into account throughout the manuscript in the revised version.

**27) P8L7-8, this should be mentioned in the study site.**

Indeed, this sentence is more descriptive and should be moved to section 3.1 where a small description of the subcatchments is done.

**28) P9: Please reconsider the presentation in this figure. You leave it to the reader what might be of importance. Furthermore, the figure is not understandable just based on the captions. Font size, line width etc. should be improved.**

Thank you for the comment. All figures will be improved in the revised version and careful revision will be made to this specific figure to pinpoint only the important results.

**29) P11: completely revise**

The intention with this figure was to display how the chemical signature of surface water within each subcatchment changes when they confluence with other tributaries. However, we will revise this figure to display clearer results.

**30) P12L2-3: You refer to event and events. How many?**

We have revised the sentence to clarify that the analysis is done with one rainfall event: *"The sampling during the rainfall event corresponds to low-medium intensity rain and it was considered as representative for rainfall-runoff evaluation. The duration of the event was 12 hours with a maximum intensity of 0.3 mm/h and an average intensity of 0.18 mm/h."*

**31) P12L2: 2mm rain is not an intensity. Was it just one event that had 2mm? These lines are confusing, I cannot figure out what was done and sampled (see suggestion about tables earlier).**

We agree with the comment. We will consider rewriting this paragraph for clarification. Tables will be added in the Annex section.

**32) P12L2ff. You do not mention the LMWL here. How does the LMWL relate to the local conditions? How many km away? On what elevation? Etc.**

We will include a small description of the Izobamba station where the LMWL was calculated. The information of the local conditions and the difference with the Izobamba station was given above, please refer to comment #4.

**33) P13: The mixing plot and hydrograph separation section should be merged.**

We will consider how to merge these two analyses in the same section for the revised version of the manuscript.

**34) P14: It is not clear here what end-members are used, but a merger with the mixing diagram section will help. Also consider presenting this information in the methods (which may not work, in case the mixing diagram was used to determine the end members, so just results in that case).**

As mentioned in the previous comment, we will merge these analyses to clarify how from the end member analysis, we identified the suitable tracers and end members for the hydrograph separation.

**35) P16L9ff. The discussion here needs some more work. You need to really make the point how your results improved both, the understanding of runoff generation in the catchment beyond previous understanding, and how this makes the work relevant for the same landscape at other places, and how the results compare to other researcher's work. The latter is needed to show the importance of the results for the community, you can close the story that you opened at the end of the introduction, where you should state (earlier comment) why this work is needed.**

We recognized the need to improve the discussion in the revised version of the manuscript and to find more comparable studies to highlight the contribution of our research for the scientific community. We believe that although no final conclusions can be withdrawn from the limited amount of data, it is important to publish the results obtained that contribute to an overall understanding of the runoff generation in this combined glacier and páramo catchment.

Annex

**Table A-1.** Location of the water samples taken from four different runoff source (Ice, Prec = precipitation, SW = springwater and Surf = surface water) and main characteristics and chemical concentrations (EC = electrical conductivity [µS/cm], $SiO_2$ [mg/l], $Cl^-$ [mg/l], $SO_4^{2-}$ [mg/l], $Na^+$ [m/l], $Mg^{2+}$ [mg/l], $K^+$ [mg/l], $Ca^{2+}$ [mg/l], $\delta^2H$ [‰], $\delta^{18}O$ [‰]) taken during July 2014. The UTM coordinates (WGS84) of the area are Zone 17M North and East, Dist = distance to the outlet, Subcat = subcatchment, Cat = catchment and Geol = geological background (Ch = Chacana volcanic rocks, Hi = Hialina lava, LaRo = Lahar Rojo, Lapl = Lavas Pleistocene, Ti = Tillite late ice age). NA = samples are not available.

| ID | Source | N (m) | E (m) | Elev. | Dist. | EC | $SiO_2$ | $Cl^-$ | $SO_4^{2-}$ | $Na^+$ | $Mg^{2+}$ | $K^+$ | $Ca^{2+}$ | $\delta^2H$ | $\delta^{18}O$ | Subcat. | Cat. | Geol. |
|---|---|---|---|---|---|---|---|---|---|---|---|---|---|---|---|---|---|---|
| 10 | Ice | 9945370 | 816341 | 4736 | 7300 | NA | NA | 2.9 | 0.6 | 1.0 | 0.1 | 1.0 | 2.2 | -112.8 | -14.4 | 23 | 2 | - |
| 20 | Ice | 9945370 | 816341 | 4736 | 7300 | NA | NA | 1.3 | 0.4 | 1.0 | 0.0 | 0.1 | 1.0 | -111.6 | -14.2 | 23 | 2 | - |
| 30 | Ice | 9945370 | 816341 | 4736 | 7300 | NA | NA | 1.5 | 0.4 | 1.0 | 0.0 | 0.0 | 1.0 | -113.6 | -14.6 | 23 | 2 | - |
| P_01 | Prec | 9943248 | 810185 | 4060 | 1 | NA | 0.7 | 3.8 | 2.2 | 1.7 | 0.3 | 1.1 | 6.7 | -59.5 | -8.5 | 1 | 1 | - |
| P_02 | Prec | 9943248 | 810185 | 4060 | 1 | 7.9 | 0.9 | 1.3 | 0.7 | 1.0 | 0.0 | 0.1 | 1.4 | -45.4 | -5.7 | 1 | 1 | - |
| P_03 | Prec | 9943248 | 810185 | 4060 | 1 | 10.0 | 1.1 | 1.6 | 0.7 | 1.0 | 0.0 | 0.1 | 1.1 | -42.4 | -3.9 | 1 | 1 | - |
| P_04 | Prec | 9943248 | 810185 | 4060 | 1 | 35.4 | 0.5 | 2.6 | 1.1 | 1.2 | 0.1 | 0.9 | 2.0 | -58.4 | -5.7 | 1 | 1 | - |
| T_S1_01 | SW | 9943424 | 810258 | 4047 | 320 | 144.5 | 57.8 | 1.9 | 10.3 | 8.7 | 4.0 | 3.0 | 7.3 | -111.0 | -13.4 | 1 | 1 | Hi |
| T_S1_20 | SW | 9943451 | 812460 | 4166 | 2850 | 70.0 | NA | NA | NA | NA | NA | NA | NA | -89.1 | -11.9 | 7 | 5 | Ch |
| T_S1_26 | SW | 9943471 | 813147 | 4206 | 3600 | 116.2 | NA | NA | NA | NA | NA | NA | NA | -85.7 | -11.9 | 9 | 5 | Ti |
| T_S1_22 | SW | 9943489 | 812029 | 4135 | 2500 | NA | NA | NA | NA | NA | NA | NA | NA | -98.9 | -12.4 | 7 | 5 | LaPl |
| T_S1_25 | SW | 9943502 | 812928 | 4191 | 3370 | NA | NA | NA | NA | NA | NA | NA | NA | -90.4 | -12.7 | 9 | 5 | LaPl |
| T_S1_21 | SW | 9943521 | 812319 | 4146 | 2800 | 116.3 | NA | NA | NA | NA | NA | NA | NA | -99.6 | -13.6 | 7 | 5 | Ti |
| T_S1_16 | SW | 9943524 | 811802 | 4127 | 3670 | NA | NA | NA | NA | NA | NA | NA | NA | -89.5 | -11.9 | 5 | 5 | LaPl |
| T_S1_15 | SW | 9943526 | 811662 | 4122 | 3675 | 140.9 | NA | NA | NA | NA | NA | NA | NA | -105.2 | -14.4 | 5 | 5 | LaPl |
| T_S1_14 | SW | 9943543 | 811655 | 4121 | 2100 | 137.7 | NA | NA | NA | NA | NA | NA | NA | -106.7 | -14.6 | 5 | 5 | LaPl |
| T_S1_17 | SW | 9943547 | 812001 | 4134 | 2540 | 94.5 | 55.8 | NA | NA | 6.3 | 4.2 | 3.1 | 6.0 | -105.3 | -13.5 | 6 | 5 | Ch |
| T_S1_09 | SW | 9943576 | 811372 | 4109 | 2020 | 114.4 | NA | NA | NA | NA | NA | NA | NA | -88.4 | -12.5 | 5 | 5 | LaPl |
| T_S1_06 | SW | 9943599 | 811130 | 4114 | 1550 | 275.0 | 69.5 | 6.8 | 19.9 | 13.8 | 7.5 | 5.8 | 14.3 | -99.6 | -13.6 | 31 | 3 | LaRo |
| T_S1_02 | SW | 9943639 | 810532 | 4077 | 817 | 159.2 | NA | NA | NA | NA | NA | NA | NA | -109.4 | -14.6 | 1 | 1 | Hi |
| T_S1_03 | SW | 9943708 | 810640 | 4079 | 1005 | 140.0 | NA | NA | NA | NA | NA | NA | NA | -109.9 | -14.7 | 1 | 1 | Hi |
| T_S1_13 | SW | 9943739 | 810657 | 4083 | 1050 | 151.0 | 57.6 | 2.3 | 14.8 | 8.6 | 3.8 | 2.9 | 6.6 | -110.5 | -13.7 | 1 | 1 | Hi |

| ID | Source | N (m) | E (m) | Elev. | Dist. | EC | $SiO_2$ | $Cl^-$ | $SO_4^{2-}$ | $Na^+$ | $Mg^{2+}$ | $K^+$ | $Ca^{2+}$ | $\delta^2H$ | $\delta^{18}O$ | Subcat. | Cat. | Geol. |
|---|---|---|---|---|---|---|---|---|---|---|---|---|---|---|---|---|---|---|
| T_S1_04 | SW | 9943741 | 810709 | 4080 | 1113 | 330.0 | NA | NA | NA | NA | NA | NA | NA | -106.0 | -14.1 | 1 | 1 | LaRo |
| T_S1_29 | SW | 9943790 | 814275 | 4325 | 4850 | 86.6 | 49.9 | NA | NA | NA | NA | NA | NA | -98.6 | -13.1 | 9 | 5 | Ti |
| T_S1_12 | SW | 9943797 | 810758 | 4083 | 1100 | 339.0 | NA | NA | NA | NA | NA | NA | NA | -112.0 | -13.0 | 1 | 1 | LaRo |
| T_S1_27 | SW | 9943812 | 814257 | 4316 | 4780 | 99.1 | NA | 1.8 | 0.7 | 3.7 | 3.0 | 2.4 | 5.1 | -101.6 | -12.6 | 9 | 5 | Ti |
| T_S1_28 | SW | 9943825 | 814281 | 4319 | 4800 | 110.8 | 46.6 | NA | NA | NA | NA | NA | NA | -93.6 | -12.6 | 9 | 5 | Ti |
| T_S1_05 | SW | 9943878 | 811156 | 4100 | 1455 | 280.0 | 64.5 | 4.9 | 35.7 | 15.6 | 7.1 | 4.4 | 12.3 | -106.6 | -14.6 | 22 | 2 | LaRo |
| T_S1_31 | SW | 9943930 | 814545 | 4341 | 5015 | NA | NA | 1.5 | 1.4 | NA | NA | NA | NA | -94.9 | -11.6 | 9 | 5 | Ti |
| T_S1_32 | SW | 9943934 | 814574 | 4345 | 5060 | 59.0 | NA | NA | NA | NA | NA | NA | NA | -95.8 | -12.3 | 9 | 5 | Ch |
| T_S1_30 | SW | 9943935 | 814517 | 4334 | 4975 | 77.7 | NA | NA | NA | NA | NA | NA | NA | -101.6 | -13.1 | 9 | 5 | Ch |
| T_S1_33 | SW | 9943973 | 814685 | 4354 | 5170 | 58.8 | NA | NA | NA | NA | NA | NA | NA | -94.1 | -12.8 | 9 | 5 | Ch |
| T_S1_11 | SW | 9943975 | 811328 | 4103 | 1790 | 298.0 | 60.8 | 5.2 | 38.5 | 18.8 | 8.5 | 5.0 | 12.9 | -108.3 | -14.3 | 22 | 2 | LaRo |
| T_S1_34 | SW | 9943985 | 814740 | 4356 | 5200 | 67.0 | NA | NA | NA | NA | NA | NA | NA | -98.6 | -12.9 | 9 | 5 | Ch |
| T_S1_35 | SW | 9944017 | 814842 | 4369 | 5400 | 64.7 | NA | NA | NA | NA | NA | NA | NA | -97.7 | -13.2 | 9 | 5 | Ch |
| S1_49 | SW | 9944045 | 813496 | 4246 | 4150 | 75.5 | NA | 1.6 | 0.5 | 4.4 | 2.0 | 2.0 | 5.2 | -84.1 | -10.6 | 8 | 5 | Ti |
| T_S1_36 | SW | 9944054 | 814937 | 4382 | 5450 | 53.1 | NA | NA | NA | NA | NA | NA | NA | -100.4 | -12.8 | 9 | 5 | Ch |
| T_S3_14 | SW | 9944054 | 811454 | 4110 | 1950 | NA | 66.1 | NA | NA | NA | NA | NA | NA | -105.5 | -14.6 | 22 | 2 | Ch |
| T_S1_37 | SW | 9944066 | 814948 | 4386 | 5475 | 67.8 | NA | NA | NA | NA | NA | NA | NA | -105.2 | -13.7 | 9 | 5 | Ch |
| T_S3_13 | SW | 9944099 | 811526 | 4113 | 2100 | NA | NA | NA | NA | NA | NA | NA | NA | -107.2 | -14.8 | 22 | 2 | Ch |
| T_S1_38 | SW | 9944100 | 815040 | 4389 | 5530 | 69.5 | NA | NA | NA | NA | NA | NA | NA | -98.1 | -14.4 | 9 | 5 | Ch |
| T_S3_12 | SW | 9944108 | 811526 | 4113 | 2170 | 112.3 | NA | NA | NA | NA | NA | NA | NA | NA | NA | 22 | 2 | Ch |
| T_S3_11 | SW | 9944111 | 811570 | 4117 | 5980 | 103.5 | NA | NA | NA | NA | NA | NA | NA | -102.2 | -13.4 | 22 | 2 | Ch |
| T_S3_10 | SW | 9944237 | 811748 | 4129 | 2370 | 91.5 | 64.2 | 2.1 | 10.0 | 6.4 | 3.3 | 3.4 | 7.4 | -106.5 | -13.4 | 22 | 2 | Ch |
| T_S3_09 | SW | 9944239 | 811757 | 4129 | 4920 | 83.6 | NA | NA | NA | NA | NA | NA | NA | -107.3 | -13.2 | 22 | 2 | Ch |
| T_S3_02 | SW | 9944265 | 812357 | 4179 | 7100 | 63.2 | NA | NA | NA | NA | NA | NA | NA | -105.7 | -13.4 | 41 | 4 | Ch |
| T_S3_01 | SW | 9944332 | 812494 | 4190 | 6750 | 71.5 | NA | NA | NA | NA | NA | NA | NA | -104.7 | -13.6 | 41 | 4 | Ch |
| T_S3_08 | SW | 9944362 | 811920 | 4149 | 2590 | 97.6 | NA | NA | NA | NA | NA | NA | NA | -102.3 | -13.8 | 22 | 2 | Ch |
| T_S3_04 | SW | 9944462 | 812125 | 4171 | 3800 | 85.2 | 67.9 | 1.9 | 4.8 | 6.2 | 3.4 | 3.6 | 7.8 | -104.9 | -13.7 | 22 | 2 | Ch |
| T_S3_03 | SW | 9944477 | 812033 | 4159 | 2550 | 90.1 | 69.6 | 2.0 | 7.5 | 6.3 | 3.1 | 3.4 | 7.6 | -102.7 | -13.4 | 22 | 2 | Ch |
| T_S3_05 | SW | 9944521 | 812189 | 4176 | 3900 | 128.1 | 62.2 | 1.8 | 35.0 | 6.1 | 3.7 | 3.4 | 9.2 | -102.7 | -12.9 | 22 | 2 | LaPl |
| T_S3_06 | SW | 9944576 | 812602 | 4209 | 2550 | 67.1 | NA | NA | NA | NA | NA | NA | NA | -95.5 | -12.0 | 22 | 2 | Ch |
| T_S3_07 | SW | 9944602 | 813584 | 4274 | 2900 | 7.4 | NA | NA | NA | NA | NA | NA | NA | -35.9 | -4.2 | 22 | 2 | Ti |

| ID | Source | N (m) | E (m) | Elev. | Dist. | EC | $SiO_2$ | $Cl^-$ | $SO_4^{2-}$ | $Na^+$ | $Mg^{2+}$ | $K^+$ | $Ca^{2+}$ | $\delta^2H$ | $\delta^{18}O$ | Subcat. | Cat. | Geol. |
|---|---|---|---|---|---|---|---|---|---|---|---|---|---|---|---|---|---|---|
| S1_01 | Surf | 9943252 | 810178 | 4044 | 100 | 127.2 | 44.8 | 2.6 | 9.4 | 7.0 | 4.0 | 2.4 | 5.9 | -98.3 | -13.1 | 1 | 1 | - |
| S1_02 | Surf | 9943306 | 810262 | 4046 | 200 | 122.8 | NA | NA | NA | NA | NA | NA | NA | -101.3 | -12.1 | 1 | 1 | - |
| S1_03 | Surf | 9943396 | 810158 | 4047 | 300 | 124.4 | 45.5 | 2.3 | 7.7 | 7.4 | 4.1 | 2.6 | 6.4 | -101.7 | -12.0 | 1 | 1 | - |
| S1_04 | Surf | 9943443 | 810304 | 4053 | 400 | 112.5 | NA | NA | NA | NA | NA | NA | NA | -98.7 | -12.1 | 1 | 1 | - |
| S1_25 | Surf | 9943452 | 812467 | 4166 | 2950 | 36.3 | NA | NA | NA | NA | NA | NA | NA | -95.6 | -13.2 | 7 | 5 | - |
| S1_05 | Surf | 9943467 | 810390 | 4066 | 500 | 116.3 | 44.5 | 2.7 | 9.7 | 7.1 | 3.8 | 2.5 | 5.9 | -98.6 | -11.9 | 1 | 1 | - |
| T_S1_10 | Surf | 9943470 | 811664 | 4122 | 2240 | 72.0 | 38.0 | 2.2 | 2.1 | 2.5 | 2.2 | 1.5 | 3.6 | -98.6 | -12.6 | 7 | 5 | - |
| S1_27 | Surf | 9943471 | 812000 | 4133 | 2480 | 45.4 | NA | NA | NA | NA | NA | NA | NA | -97.5 | -12.5 | 7 | 5 | - |
| S1_24 | Surf | 9943477 | 812678 | 4179 | 3200 | 59.7 | 42.5 | 1.6 | 0.5 | 3.4 | 2.8 | 2.4 | 4.7 | -82.5 | -10.9 | 8 | 5 | - |
| S1_28 | Surf | 9943478 | 812864 | 4187 | 3300 | 57.2 | 37.6 | 1.7 | 0.9 | 2.4 | 1.5 | 1.6 | 3.0 | -96.6 | -13.4 | 9 | 5 | - |
| S1_29 | Surf | 9943481 | 813062 | 4200 | 3500 | 54.7 | NA | NA | NA | NA | NA | NA | NA | -95.3 | -13.4 | 9 | 5 | - |
| S1_06 | Surf | 9943505 | 810474 | 4076 | 600 | 110.9 | NA | NA | NA | NA | NA | NA | NA | -98.3 | -12.2 | 1 | 1 | - |
| S1_26 | Surf | 9943507 | 812253 | 4143 | 2750 | 43.7 | 40.3 | 1.6 | 0.8 | 2.0 | 1.5 | 1.3 | 3.1 | -98.2 | -12.5 | 7 | 5 | - |
| T_S1_19 | Surf | 9943514 | 812666 | 4179 | 2700 | 68.2 | NA | NA | NA | NA | NA | NA | NA | -82.5 | -10.9 | 8 | 5 | - |
| S1_18 | Surf | 9943529 | 811891 | 4129 | 2500 | 48.9 | NA | NA | NA | NA | NA | NA | NA | -104.5 | -13.9 | 6 | 5 | - |
| S1_07 | Surf | 9943548 | 810515 | 4078 | 700 | 124.5 | 46.2 | 2.7 | 9.9 | 7.8 | 4.2 | 2.8 | 6.3 | -100.0 | -12.5 | 1 | 1 | - |
| S1_16 | Surf | 9943548 | 811455 | 4112 | 2190 | 127.7 | NA | NA | NA | NA | NA | NA | NA | -102.4 | -14.2 | 5 | 5 | - |
| B_1 | Surf | 9943548 | 811455 | 4116 | 1850 | 128.7 | 38.8 | 2.4 | 6.3 | 5.5 | 2.8 | 2.0 | 5.1 | NA | NA | 31 | 3 | - |
| S1_19 | Surf | 9943557 | 812079 | 4139 | 2700 | 47.7 | 48.5 | 1.8 | 1.2 | 3.9 | 1.7 | 2.2 | 4.1 | -108.6 | -13.8 | 6 | 5 | - |
| S1_17 | Surf | 9943558 | 811675 | 4122 | 2300 | 52.9 | 45.6 | 1.9 | 1.4 | 4.1 | 2.1 | 2.1 | 3.9 | -102.3 | -13.5 | 6 | 5 | - |
| A_1 | Surf | 9943562 | 811460 | 4116 | 2000 | 254.8 | 57.8 | 6.4 | 15.7 | 10.5 | 6.0 | 4.9 | 18.5 | NA | NA | 32 | 3 | - |
| S1_23 | Surf | 9943566 | 812862 | 4189 | 3400 | 50.1 | NA | NA | NA | NA | NA | NA | NA | -82.8 | -11.5 | 8 | 5 | - |
| S1_30 | Surf | 9943569 | 813246 | 4214 | 3700 | 54.2 | 35.5 | 1.3 | 1.1 | 2.7 | 1.9 | 1.8 | 4.2 | -96.8 | -13.5 | 9 | 5 | - |
| S1_08 | Surf | 9943621 | 810526 | 4078 | 800 | 117.9 | NA | NA | NA | NA | NA | NA | NA | -99.1 | -12.5 | 1 | 1 | - |
| T_S1_23 | Surf | 9943624 | 812177 | 4149 | 2670 | 44.1 | NA | NA | NA | NA | NA | NA | NA | -107.8 | -13.7 | 6 | 5 | - |
| S1_15 | Surf | 9943630 | 811327 | 4107 | 1900 | 106.1 | 45.6 | 2.3 | 6.0 | 5.7 | 3.4 | 2.3 | 5.0 | -99.8 | -13.3 | 5 | 5 | - |
| T_S1_24 | Surf | 9943638 | 812201 | 4155 | 2680 | 43.5 | 47.5 | 1.9 | 1.3 | 2.8 | 1.3 | 1.6 | 2.8 | -108.3 | -13.6 | 6 | 5 | - |
| S1_22 | Surf | 9943661 | 813059 | 4204 | 3600 | 47.3 | 44.3 | 1.5 | 0.4 | 4.2 | 2.9 | 2.8 | 5.8 | -84.6 | -11.2 | 8 | 5 | - |
| S1_31 | Surf | 9943669 | 813421 | 4230 | 3900 | 52.3 | NA | NA | NA | NA | NA | NA | NA | -95.3 | -13.1 | 9 | 5 | - |
| 3_2C | Surf | 9943673 | 811237 | 4103 | 1620 | 142.0 | 42.3 | 1.8 | 3.9 | 6.7 | 3.4 | 2.3 | 5.4 | NA | NA | 31 | 3 | - |
| S3_17 | Surf | 9943675 | 811337 | 4108 | 2020 | 75.2 | NA | NA | NA | NA | NA | NA | NA | -103.1 | -12.4 | 41 | 4 | - |

| ID | Source | N (m) | E (m) | Elev. | Dist. | EC | SiO$_2$ | Cl$^-$ | SO$_4^{2-}$ | Na$^+$ | Mg$^{2+}$ | K$^+$ | Ca$^{2+}$ | $\delta^2$H | $\delta^{18}$O | Subcat. | Cat. | Geol. |
|---|---|---|---|---|---|---|---|---|---|---|---|---|---|---|---|---|---|---|
| S1_32 | Surf | 9943689 | 813619 | 4244 | 4100 | 52.8 | 35.8 | 1.5 | 1.0 | 2.9 | 1.8 | 1.9 | 4.1 | -99.2 | -13.0 | 9 | 5 | - |
| S1_09 | Surf | 9943690 | 810581 | 4078 | 900 | 120.0 | 45.5 | 2.8 | 10.1 | 7.0 | 3.5 | 2.6 | 5.6 | -100.4 | -12.4 | 1 | 1 | - |
| T_S1_08 | Surf | 9943703 | 811488 | 4116 | 1120 | 93.5 | 49.4 | 1.5 | 2.4 | 4.8 | 3.0 | 2.5 | 4.7 | -98.9 | -13.4 | 41 | 4 | - |
| S1_10 | Surf | 9943706 | 810637 | 4079 | 1000 | 130.0 | NA | NA | NA | NA | NA | NA | NA | -99.5 | -13.3 | 1 | 1 | - |
| S3_16 | Surf | 9943711 | 811540 | 4119 | 950 | 69.1 | 50.5 | 1.7 | 2.4 | 5.3 | 2.8 | 2.8 | 5.0 | -103.2 | -12.7 | 41 | 4 | - |
| C_1 | Surf | 9943717 | 811140 | 4114 | 1840 | 131.6 | 39.7 | 2.9 | 9.3 | 3.8 | 1.9 | 1.5 | 3.2 | NA | NA | 31 | 3 | - |
| S1_14 | Surf | 9943724 | 811178 | 4100 | 1500 | 144.9 | 42.7 | 3.2 | 6.5 | 6.7 | 3.4 | 2.3 | 8.1 | -101.9 | -13.5 | 31 | 3 | - |
| S3_15 | Surf | 9943733 | 811602 | 4123 | 1220 | 90.1 | 49.6 | 1.6 | 0.6 | 6.9 | 5.7 | 3.7 | 6.8 | -92.0 | -11.4 | 41 | 4 | - |
| S1_11 | Surf | 9943735 | 810706 | 4082 | 1100 | 110.0 | 45.3 | 2.9 | 9.5 | 6.9 | 3.7 | 2.6 | 5.9 | -101.0 | -12.2 | 1 | 1 | - |
| S1_12 | Surf | 9943746 | 810801 | 4088 | 1200 | 120.0 | NA | NA | NA | NA | NA | NA | NA | -99.6 | -13.1 | 1 | 1 | - |
| S1_33 | Surf | 9943756 | 813817 | 4259 | 4300 | 52.0 | NA | NA | NA | NA | NA | NA | NA | -98.7 | -12.8 | 9 | 5 | - |
| S1_13 | Surf | 9943767 | 810978 | 4092 | 1300 | 134.9 | 46.8 | 3.0 | 10.7 | 7.4 | 4.0 | 2.7 | 6.4 | -102.6 | -13.4 | 1 | 1 | - |
| 21_1_2C | Surf | 9943783 | 811014 | 4094 | 1345 | 247.2 | 59.7 | 1.5 | 19.1 | 11.4 | 5.9 | 3.3 | 10.5 | NA | NA | 21 | 2 | - |
| S1_34 | Surf | 9943790 | 914018 | 4281 | 4500 | 49.1 | 34.5 | 1.9 | 2.2 | 2.2 | 1.4 | 1.5 | 2.9 | -105.9 | -13.7 | 9 | 5 | - |
| S1_35 | Surf | 9943814 | 814221 | 4314 | 4700 | 50.6 | NA | NA | NA | NA | NA | NA | NA | -101.4 | -12.8 | 9 | 5 | - |
| S1_21 | Surf | 9943817 | 813187 | 4225 | 3800 | 46.3 | NA | NA | NA | NA | NA | NA | NA | -82.4 | -11.4 | 8 | 5 | - |
| S3_14 | Surf | 9943854 | 811772 | 4133 | 1545 | 91.5 | NA | NA | NA | NA | NA | NA | NA | -95.4 | -11.8 | 41 | 4 | - |
| S1_36 | Surf | 9943871 | 814412 | 4324 | 4900 | 53.3 | 35.9 | 1.4 | 0.7 | 2.5 | 1.5 | 1.6 | 3.4 | -103.6 | -13.1 | 9 | 5 | - |
| 22_1_2C | Surf | 9943878 | 811156 | 4050 | 1445 | 280.0 | 64.5 | 4.9 | NA | 15.6 | 7.1 | 4.4 | 12.3 | NA | NA | 22 | 2 | - |
| 21_2_2C | Surf | 9943894 | 811126 | 4100 | 1420 | 202.0 | 47.9 | 0.8 | 11.8 | 9.8 | 4.9 | 2.8 | 8.9 | NA | NA | 21 | 2 | - |
| S2_28 | Surf | 9943926 | 811107 | 4101 | 1500 | 11.8 | 10.6 | 1.3 | 0.8 | 1.0 | 0.3 | 0.4 | 1.2 | -89.9 | -12.5 | 23 | 2 | - |
| S1_20 | Surf | 9943934 | 813378 | 4236 | 4000 | 42.5 | 40.7 | 1.6 | 0.5 | 3.3 | 2.0 | 2.2 | 4.4 | -80.9 | -11.3 | 8 | 5 | - |
| S1_37 | Surf | 9943957 | 814637 | 4347 | 5100 | 49.9 | NA | NA | NA | NA | NA | NA | NA | -99.5 | -13.4 | 9 | 5 | - |
| S1_38 | Surf | 9944019 | 814821 | 4365 | 5300 | 49.0 | 32.7 | 1.6 | 1.3 | 1.5 | 0.9 | 1.0 | 3.7 | -96.7 | -13.7 | 9 | 5 | - |
| S3_13 | Surf | 9944046 | 811889 | 4146 | 1990 | 87.7 | 52.4 | 1.6 | 0.9 | 5.8 | 4.7 | 3.2 | 5.7 | -95.0 | -11.8 | 41 | 4 | - |
| S1_39 | Surf | 9944090 | 815015 | 4386 | 5500 | 46.1 | NA | NA | NA | NA | NA | NA | NA | -96.0 | -13.5 | 9 | 5 | - |
| S2_29 | Surf | 9944102 | 811177 | 4127 | 1700 | 9.4 | 10.5 | 1.6 | 1.0 | 1.3 | 0.3 | 0.5 | 1.5 | -90.0 | -12.4 | 23 | 2 | - |
| S3_12 | Surf | 9944139 | 812070 | 4161 | 2190 | 71.3 | NA | NA | NA | NA | NA | NA | NA | -104.9 | -13.2 | 41 | 4 | - |
| S1_40 | Surf | 9944153 | 815208 | 4410 | 5700 | 41.1 | 27.2 | 1.4 | 1.6 | 1.5 | 1.0 | 1.0 | 2.6 | -96.5 | -13.7 | 9 | 5 | - |
| S3_11 | Surf | 9944207 | 812233 | 4172 | 2370 | 73.4 | 49.5 | 1.9 | 0.6 | 3.9 | 2.7 | 3.1 | 5.0 | -98.4 | -12.3 | 41 | 4 | - |
| S2_27 | Surf | 9944207 | 811300 | 4138 | 1900 | 10.2 | NA | NA | NA | NA | NA | NA | NA | -91.5 | -12.3 | 23 | 2 | - |

| ID | Source | N (m) | E (m) | Elev. | Dist. | EC | $SiO_2$ | $Cl^-$ | $SO_4^{2-}$ | $Na^+$ | $Mg^{2+}$ | $K^+$ | $Ca^{2+}$ | $\delta^2H$ | $\delta^{18}O$ | Subcat. | Cat. | Geol. |
|---|---|---|---|---|---|---|---|---|---|---|---|---|---|---|---|---|---|---|
| S1_41 | Surf | 9944209 | 815393 | 4431 | 5900 | 33.9 | NA | NA | NA | NA | NA | NA | NA | -99.8 | -13.1 | 9 | 5 | - |
| S1_46 | Surf | 9944216 | 816351 | 4596 | 7000 | 23.5 | 11.4 | 2.1 | 3.2 | 1.2 | 0.5 | 0.7 | 2.3 | -100.8 | -13.7 | 9 | 5 | - |
| S1_45 | Surf | 9944223 | 816189 | 4567 | 6800 | 24.8 | NA | NA | NA | NA | NA | NA | NA | -99.8 | -12.5 | 9 | 5 | - |
| S1_44 | Surf | 9944233 | 816003 | 4538 | 6600 | 26.5 | 12.5 | 2.1 | 2.7 | 1.3 | 0.6 | 0.8 | 2.4 | -96.8 | -13.0 | 9 | 5 | - |
| S1_47 | Surf | 9944273 | 816511 | 4634 | 7200 | 22.9 | NA | NA | NA | NA | NA | NA | NA | -98.9 | -13.4 | 9 | 5 | - |
| S1_42 | Surf | 9944323 | 815567 | 4462 | 6100 | 32.5 | 19.8 | 1.3 | 2.0 | 1.4 | 0.8 | 1.0 | 2.7 | -97.7 | -13.9 | 9 | 5 | - |
| S3_08 | Surf | 9944327 | 812892 | 4221 | 7000 | 41.0 | NA | NA | NA | NA | NA | NA | NA | -95.5 | -11.6 | 41 | 4 | - |
| T_S1_39 | Surf | 9944328 | 815945 | 4522 | 6550 | 39.2 | 25.7 | NA | NA | 2.2 | 0.9 | 1.3 | 2.8 | -108.0 | -13.7 | 9 | 5 | - |
| S3_09 | Surf | 9944331 | 812703 | 4207 | 6600 | 39.2 | 48.4 | 1.7 | 1.1 | 2.7 | 1.4 | 1.7 | 2.2 | -100.3 | -12.3 | 41 | 4 | - |
| S3_10 | Surf | 9944331 | 812495 | 4190 | 7300 | 70.5 | NA | NA | NA | NA | NA | NA | NA | -102.5 | -13.2 | 41 | 4 | - |
| S1_43 | Surf | 9944356 | 815746 | 4492 | 6300 | 31.8 | NA | NA | NA | NA | NA | NA | NA | -97.3 | -13.7 | 9 | 5 | - |
| S1_48 | Surf | 9944382 | 816692 | 4673 | 7400 | 21.3 | 7.5 | 1.4 | 3.6 | 1.0 | 0.6 | 0.6 | 3.1 | -100.9 | -12.8 | 9 | 5 | - |
| S2_26 | Surf | 9944402 | 811469 | 4148 | 2100 | 10.3 | 6.2 | 1.3 | 1.0 | 1.8 | 0.3 | 0.6 | 1.9 | -90.7 | -12.4 | 23 | 2 | - |
| S3_07 | Surf | 9944403 | 813078 | 4236 | 3700 | 55.4 | 46.2 | 1.9 | 1.3 | 2.1 | 1.8 | 1.2 | 2.2 | -100.6 | -12.2 | 41 | 4 | - |
| S3_06 | Surf | 9944505 | 813280 | 4257 | 3900 | 64.7 | NA | NA | NA | NA | NA | NA | NA | -101.4 | -12.9 | 43 | 4 | - |
| S3_18 | Surf | 9944534 | 813410 | 4268 | 4030 | 55.1 | 44.6 | 1.6 | 0.6 | 3.3 | 3.2 | 2.0 | 4.3 | -92.4 | -12.1 | 42 | 4 | - |
| S3_19 | Surf | 9944594 | 813565 | 4286 | 4230 | 32.9 | NA | NA | NA | NA | NA | NA | NA | -99.7 | -13.1 | 42 | 4 | - |
| S2_25 | Surf | 9944600 | 811518 | 4157 | 2300 | 10.8 | NA | NA | NA | NA | NA | NA | NA | -88.7 | -12.3 | 23 | 2 | - |
| S3_20 | Surf | 9944666 | 813704 | 4297 | 4430 | 51.2 | 62.8 | 2.0 | 1.2 | 4.1 | 1.8 | 2.8 | 4.3 | -97.0 | -15.7 | 42 | 4 | - |
| S3_01 | Surf | 9944677 | 813367 | 4281 | 4100 | 68.7 | 54.1 | 1.6 | 1.7 | 2.3 | 2.7 | 1.4 | 2.8 | -105.6 | -12.9 | 43 | 4 | - |
| S3_21 | Surf | 9944713 | 813880 | 4314 | 4630 | 30.7 | NA | NA | NA | NA | NA | NA | NA | -103.7 | -12.7 | 42 | 4 | - |
| S3_02 | Surf | 9944766 | 813536 | 4303 | 4300 | 73.8 | NA | NA | NA | NA | NA | NA | NA | -104.8 | -12.9 | 43 | 4 | - |
| S2_24 | Surf | 9944809 | 811679 | 4170 | 2500 | 9.8 | 11.5 | 1.5 | 0.9 | 1.4 | 0.2 | 0.4 | 1.1 | -87.9 | -12.7 | 23 | 2 | - |
| S3_22 | Surf | 9944829 | 814097 | 4343 | 4830 | 45.8 | 53.2 | 1.8 | 1.6 | 4.0 | 2.0 | 2.6 | 4.9 | -110.8 | -14.0 | 42 | 4 | - |
| S3_03 | Surf | 9944918 | 813657 | 4317 | 4500 | 69.5 | 58.3 | 1.7 | 1.4 | 3.7 | 3.1 | 2.6 | 4.0 | -105.7 | -13.3 | 43 | 4 | - |
| S2_23 | Surf | 9944947 | 811791 | 4179 | 2700 | 9.5 | NA | NA | NA | NA | NA | NA | NA | -88.5 | -12.9 | 23 | 2 | - |
| S2_01 | Surf | 9945081 | 811901 | 4188 | 2900 | 7.6 | 12.8 | 1.2 | 0.8 | 1.0 | 0.3 | 0.3 | 1.5 | -93.0 | -13.0 | 23 | 2 | - |
| S3_04 | Surf | 9945090 | 813820 | 4341 | 4700 | 68.2 | NA | NA | NA | NA | NA | NA | NA | -107.8 | -13.8 | 43 | 4 | - |
| S2_02 | Surf | 9945155 | 812102 | 4208 | 3100 | 6.6 | NA | NA | NA | NA | NA | NA | NA | -93.5 | -13.0 | 23 | 2 | - |
| S3_05 | Surf | 9945164 | 813983 | 4370 | 4900 | 87.4 | 54.7 | 1.7 | 0.9 | 3.1 | 5.1 | 3.1 | 5.2 | -106.5 | -13.5 | 43 | 4 | - |
| S2_03 | Surf | 9945262 | 812266 | 4232 | 3300 | 6.7 | 11.2 | 1.3 | 0.9 | 1.2 | 0.2 | 0.4 | 1.4 | -95.2 | -12.9 | 23 | 2 | - |

| ID | Source | N (m) | E (m) | Elev. | Dist. | EC | $SiO_2$ | $Cl^-$ | $SO_4^{2-}$ | $Na^+$ | $Mg^{2+}$ | $K^+$ | $Ca^{2+}$ | $\delta^2H$ | $\delta^{18}O$ | Subcat. | Cat. | Geol. |
|---|---|---|---|---|---|---|---|---|---|---|---|---|---|---|---|---|---|---|
| Lake | Surf | 9945276 | 815577 | 4664 | 7050 | 6.6 | 3.9 | 1.2 | 0.7 | 1.6 | 0.2 | 0.5 | 1.0 | -87.6 | -12.6 | 23 | 2 | - |
| S2_22 | Surf | 9945276 | 815578 | 4664 | 7050 | 6.6 | 3.9 | 1.4 | 0.8 | 1.0 | 0.2 | 0.3 | 1.0 | NA | NA | 23 | 2 | - |
| S2_21 | Surf | 9945331 | 815528 | 4650 | 6900 | 6.3 | 3.7 | 1.7 | 0.8 | NA | NA | NA | NA | -94.9 | -13.4 | 23 | 2 | - |
| S2_04 | Surf | 9945351 | 812415 | 4237 | 3500 | 6.7 | NA | NA | NA | NA | NA | NA | NA | -94.1 | -12.9 | 23 | 2 | - |
| S2_05 | Surf | 9945411 | 812613 | 4260 | 3700 | 6.5 | 13.3 | 1.2 | 0.8 | 1.2 | 0.2 | 0.3 | 1.5 | -94.2 | -12.8 | 23 | 2 | - |
| S2_20 | Surf | 9945449 | 815378 | 4597 | 6700 | 6.4 | NA | NA | NA | NA | NA | NA | NA | -95.5 | -13.4 | 23 | 2 | - |
| S2_15 | Surf | 9945503 | 814384 | 4431 | 5700 | 6.9 | 5.6 | 1.2 | 0.8 | 1.3 | 0.3 | 0.4 | 1.5 | -97.4 | -13.1 | 23 | 2 | - |
| S2_19 | Surf | 9945504 | 815186 | 4552 | 6500 | 6.4 | 9.7 | 1.6 | 0.8 | 1.0 | 0.2 | 0.2 | 1.0 | -93.0 | -13.4 | 23 | 2 | - |
| S2_17 | Surf | 9945509 | 814813 | 4489 | 6100 | 6.6 | 6.3 | 1.2 | 0.8 | 1.0 | 0.2 | 0.3 | 1.5 | -97.2 | -12.9 | 23 | 2 | - |
| S2_14 | Surf | 9945510 | 814192 | 4413 | 5500 | 6.9 | NA | NA | NA | NA | NA | NA | NA | -96.9 | -13.2 | 23 | 2 | - |
| S2_18 | Surf | 9945524 | 814995 | 4521 | 6300 | 7.5 | NA | NA | NA | NA | NA | NA | NA | -95.8 | -12.8 | 23 | 2 | - |
| S2_06 | Surf | 9945534 | 812806 | 4275 | 3900 | 6.5 | NA | NA | NA | NA | NA | NA | NA | -95.0 | -12.7 | 23 | 2 | - |
| S2_16 | Surf | 9945534 | 814618 | 4460 | 5900 | 6.7 | NA | NA | NA | NA | NA | NA | NA | -96.6 | -12.9 | 23 | 2 | - |
| S2_07 | Surf | 9945599 | 813004 | 4294 | 4100 | 6.5 | 10.1 | 1.5 | 0.8 | 1.5 | 0.2 | 0.5 | 1.6 | -96.0 | -12.6 | 23 | 2 | - |
| S2_13 | Surf | 9945612 | 813989 | 4397 | 5300 | 6.9 | 9.5 | 1.2 | 0.8 | 1.2 | 0.2 | 0.4 | 1.4 | -97.2 | -13.1 | 23 | 2 | - |
| S2_12 | Surf | 9945716 | 813831 | 4370 | 5100 | 7.0 | NA | NA | NA | NA | NA | NA | NA | -94.0 | -12.4 | 23 | 2 | - |
| S2_08 | Surf | 9945758 | 813117 | 4303 | 4300 | 6.4 | NA | NA | NA | NA | NA | NA | NA | -94.0 | -12.6 | 23 | 2 | - |
| S2_11 | Surf | 9945769 | 813624 | 4353 | 4900 | 6.9 | 13.1 | 1.2 | 0.8 | NA | NA | NA | NA | -94.6 | -12.5 | 23 | 2 | - |
| S2_10 | Surf | 9945858 | 813448 | 4338 | 4700 | 6.7 | NA | NA | NA | NA | NA | NA | NA | -93.3 | -12.4 | 23 | 2 | - |
| S2_09 | Surf | 9945899 | 813229 | 4320 | 4500 | 6.5 | 11.5 | 1.3 | 0.8 | 1.1 | 0.1 | 0.3 | 1.3 | -93.8 | -12.5 | 23 | 2 | - |

**Table A2.** Statistical summary of the chemical components and stable istopes of water samples within the Los Crespos-Humboldt basin.

| | EC | $SiO_2$ | $Cl^-$ | $SO_4^{2-}$ | $Na^+$ | $Mg^{2+}$ | $K^+$ | $Ca^{2+}$ | $\delta^2 H$ | $\delta^{18}O$ |
|---|---|---|---|---|---|---|---|---|---|---|
| *Ice (n=3)* | | | | | | | | | | |
| mean | - | - | 1.9 | 0.5 | 1.0 | 0.0 | 0.3 | 1.3 | -112.7 | -14.4 |
| std | - | - | 0.9 | 0.1 | 0.0 | 0.1 | 0.5 | 0.6 | 1.0 | 0.2 |
| min | - | - | 1.3 | 0.4 | 1.0 | 0.0 | 0.0 | 1.0 | -113.6 | -14.6 |
| max | - | - | 2.9 | 0.6 | 1.0 | 0.1 | 1.0 | 2.2 | -111.6 | -14.2 |
| *Precipitation (n=4)* | | | | | | | | | | |
| mean | 17.8 | 0.8 | 2.3 | 1.2 | 1.2 | 0.1 | 0.6 | 2.8 | -51.4 | -5.9 |
| std | 15.3 | 0.3 | 1.1 | 0.7 | 0.3 | 0.1 | 0.5 | 2.6 | 8.8 | 1.9 |
| min | 7.9 | 0.5 | 1.3 | 0.7 | 1.0 | 0.0 | 0.1 | 1.1 | -59.5 | -8.5 |
| max | 35.4 | 1.1 | 3.8 | 2.2 | 1.7 | 0.3 | 1.1 | 6.7 | -42.4 | -3.9 |
| *Springwater (n=46)* | | | | | | | | | | |
| mean | 119.9 | 61.0 | 2.8 | 14.9 | 8.7 | 4.5 | 3.5 | 8.5 | -99.5 | -13.1 |
| std | 77.7 | 7.2 | 1.7 | 14.2 | 4.8 | 2.1 | 1.1 | 3.1 | 12.0 | 1.6 |
| min | 7.4 | 46.6 | 1.5 | 0.5 | 3.7 | 2.0 | 2.0 | 5.1 | -112.0 | -14.8 |
| max | 339.0 | 69.6 | 6.8 | 38.5 | 18.8 | 8.5 | 5.8 | 14.3 | -35.9 | -4.2 |
| *Surface water (n=113)* | | | | | | | | | | |
| mean | 59.6 | 34.1 | 1.9 | 3.2 | 3.8 | 2.2 | 1.8 | 4.2 | -97.5 | -12.8 |
| std | 53.9 | 18.2 | 0.9 | 4.1 | 3.0 | 1.8 | 1.1 | 2.9 | 6.0 | 0.7 |
| min | 6.3 | 3.7 | 0.8 | 0.4 | 1.0 | 0.1 | 0.2 | 1.0 | -110.8 | -15.7 |
| max | 280.0 | 64.5 | 6.4 | 19.1 | 15.6 | 7.1 | 4.9 | 18.5 | -80.9 | -10.9 |